# Stabilization of Pin1 by USP34 promotes Ubc9 isomerization and protein sumoylation in glioma stem cells

Qiuhong Zhu [1,2,7], Panpan Liang[1,2,7], Hao Meng[1,2], Fangzhen Li[1,2], Wei Miao[1,2], Cuiying Chu[1,2], Wei Wang[1,2], Dongxue Li[2,3], Cong Chen[2,4], Yu Shi [4], Xingjiang Yu[5], Yifang Ping [2,4], Chaoshi Niu[3], Hai-bo Wu[1,2], Aili Zhang [6] ✉, Xiu-wu Bian [2,4] ✉ & Wenchao Zhou [1,2] ✉

The peptidyl-prolyl cis-trans isomerase Pin1 is a pivotal therapeutic target in cancers, but the regulation of Pin1 protein stability is largely unknown. High Pin1 expression is associated with SUMO1-modified protein hypersumoylation in glioma stem cells (GSCs), but the underlying mechanisms remain elusive. Here we demonstrate that Pin1 is deubiquitinated and stabilized by USP34, which promotes isomerization of the sole SUMO E2 enzyme Ubc9, leading to SUMO1-modified hypersumoylation to support GSC maintenance. Pin1 interacts with USP34, a deubiquitinase with preferential expression and oncogenic function in GSCs. Such interaction is facilitated by Plk1-mediated phosphorylation of Pin1. Disruption of USP34 or inhibition of Plk1 promotes polyubiquitination and degradation of Pin1. Furthermore, Pin1 isomerizes Ubc9 to upregulate Ubc9 thioester formation with SUMO1, which requires CDK1-mediated phosphorylation of Ubc9. Combined inhibition of Pin1 and CDK1 with sulfopin and RO3306 most effectively suppresses orthotopic tumor growth. Our findings provide multiple molecular targets to induce Pin1 degradation and suppress hypersumoylation for cancer treatment.

Post-translational protein processing determines the properties of proteins with indispensable functions in malignant cancers such as the extremely lethal brain tumor glioblastoma (GBM). Glioma stem cells (GSCs) at the apex of tumor cell hierarchy play pivotal roles in initiation, progression, and therapeutic resistance of GBMs[1,2]. As the most intensively studied category of protein processing, post-translational modifications (PTMs) have been detected on several important proteins involved in self-renewal, pluripotency, and stress response of GSCs[3–5]. Importantly, PTM-related mechanistic studies in GSCs have revealed many inhibitory compounds as potential therapeutics[3,4]. Protein conformational changes, particularly the intrinsic conformational switch caused by prolyl cis-trans isomerization, represent another category of post-translational protein processing with essential functions in tumor biology[6]. Studying the upstream regulatory molecules as well as the downstream oncogenic substrates of prolyl isomerization in GSCs will certainly lead to the discovery of novel

[1]Department of Pathology, The First Affiliated Hospital of USTC, Division of Life Sciences and Medicine, University of Science and Technology of China, Hefei, Anhui, China. [2]Intelligent Pathology Institute, The First Affiliated Hospital of USTC, Division of Life Sciences and Medicine, University of Science and Technology of China, Hefei, Anhui, China. [3]Department of Neurosurgery, the First Affiliated Hospital of USTC, Division of Life Sciences and Medicine, University of Science and Technology of China, Hefei, China. [4]Institute of Pathology and Southwest Cancer Center, Southwest Hospital, Third Military Medical University (Army Medical University) and Key Laboratory of Tumor Immunopathology, Ministry of Education of China, Chongqing, China. [5]Department of Histology and Embryology, School of Basic Medicine, Tongji Medical College, Huazhong University of Science and Technology, Wuhan, China. [6]Department of Cell Biology, School of Life Science, Anhui Medical University, Hefei, Anhui, China. [7]These authors contributed equally: Qiuhong Zhu, Panpan Liang. ✉e-mail: ailizhang829@163.com; bianxiuwu@263.net; WZAZ@ustc.edu.cn

therapeutic targets and therefore should be a top priority for cancer research.

Prolyl cis-trans isomerization stands for the rotation of the peptidyl prolyl bonds to adopt the cis or trans conformation, which leads to a spatial change of the backbone segments and a functional alternation of the proteins[7]. This process is catalyzed by enzymes called peptidyl prolyl cis-trans isomerases (PPIases), including cyclophilins, FK506-binding proteins (FKBPs), and parvulins. Although PPIases have garnered much attention because of their roles in mTOR signaling and immunosuppression, the isomerase activity is often not necessary for their functions[6]. However, the PPIase activity is essential for the functions of peptidyl-prolyl cis-trans isomerase NIMA-interacting 1 (Pin1), which promotes malignant development of several kinds of cancers. As a member of the parvulin PPIase family, Pin1 specifically recognizes the phosphorylated serine (S) or threonine (T) followed by a proline (Pro), and isomerizes the pS/T-Pro bond effectively[6]. Through isomerizing hundreds of substrate proteins, Pin1 functions as a pivotal regulator in multiple aspects in cancers. Pin1 often regulate the protein stability or change the subcellular localizations of substrate proteins, mostly powerful oncogenes, to promote tumor growth[8,9]. Through modulating the conformation of the BRCA1-BARD1 complex, Pin1 is involved in the replication fork protections associated with cancer development[10]. By upregulating the levels of hypoxia inducible factor (HIF), Pin1 participates in potentiating tumor hypoxia responses[11]. Pin1 acts in tumor metabolic reprogramming by regulating a variety of metabolic enzymes, including fatty acid synthase (FASN), phosphoglycerate kinase 1 (PGK1) and pyruvate kinase2 (PKM2)[12–14]. Pin1 in tumor cells and cancer-associated fibroblasts (CAFs) functions to constitute an immunosuppressive microenvironment to facilitate tumor growth[15]. Finally, Pin1 is highly expressed in cancer stem cells and supports the self-renewal of these poorly differentiated tumor cells[4,9,16,17], but the detailed downstream effectors remain elusive. In summary, Pin1 is the crucial PPIase with a central role in signaling networks in cancers.

Pin1 is frequently upregulated in several human cancers, and its overexpression is correlated with poor prognosis[6,17,18]. Thus, enormous efforts have been put into the study concerning the regulation of Pin1 expression, leading to the discovery of multiple layers of mechanisms controlling Pin1 levels in cancers. Most studies have been focusing on the regulation of Pin1 at the transcriptional level. For example, the E2F family of transcription factors have been found to bind to the Pin1 promoter and activate Pin1 transcription at the downstream of oncogenic signaling[19]. A cohort of reports reveal the post-transcriptional regulation of Pin1. The Pin1 mRNA stability is reported to be inhibited by microRNAs such as miR-628-5p and miR-140-5p[20,21]. In addition, there are few investigations concerning the Pin1 protein stability at the post-translational level. Pin1 can be phosphorylated at S65 by the polo-like kinase 1 (Plk1), which inhibits Pin1 poly-ubiquitylation and elevates Pin1 protein levels[22]. Likewise, c-Jun N-terminal kinase (JNK) phosphorylates Pin1 at S115 to inhibit the mono-ubiquitination and proteasomal degradation of Pin1[23]. Despite these discoveries, our knowledge about the mechanisms underlying aberrant Pin1 expression in cancers remains very limited. In particular, little is known about the enzymes responsible for the addition or removal of the ubiquitin chain that determine the Pin1 protein stability.

The ubiquitin-proteasome system is commonly applied by cells to control the abundance of most proteins. Whereas ubiquitination usually guides proteins to degradation, the counteracting removal of ubiquitin chains by the deubiquitinases (DUBs) enhances protein stability[24,25]. Compared with approximately 600 E3 ubiquitin ligases, there are only around 100 DUBs in human, highlighting the importance of the DUBs both as critical regulators of cellular life processes and as crucial drug targets for disease treatment[24,25]. Indeed, DUBs are widely involved in tumor development. For instance, the DUB ubiquitin-specific protease 7 (USP7) reduces the ubiquitination and

degradation of MDM2 that functions as the oncogenic E3 ligase of the foremost tumor suppressor p53 in various cancers[26]. Moreover, accumulating evidences suggest that DUBs have essential roles in the maintenance of cancer stem cells. In breast cancer stem cells, USP37 affects stemness, epithelial-mesenchymal transition and cisplatin sensitivity through regulating the hedgehog (Hh) pathway[27]. In GSCs, USP13 promotes self-renewal by deubiquitinating of the stem cell transcription factor c-Myc[28]. Furthermore, in the hypoxic niches, USP33 is upregulated upon hypoxia to deubiquitinate HIF2α and support GSC maintenance[5]. However, although ubiquitination has been reported to control Pin1 stability, the potential DUB of Pin1 has never been identified.

The covalent conjugation of the small ubiquitin-like modifier (SUMO) to substrate proteins, named as protein sumoylation, has been recognized as an important PTM in tumor biology[29]. Pin1 has been reported to promote SUMO1-modified protein sumoylation that is crucial for maintaining the tumorigenic capacity of GSCs[4], but the contribution of the PPIase activity of Pin1 in the global hypersumoylation remains unknown. Pin1 may change the conformation of key components of the sumoylation machinery to exert a universal effect on sumoylation substrates. The sumoylation machinery is composed of SUMO proteins, the SUMO-conjugating enzymes and the SUMO-deconjugating enzymes. The SUMO-conjugating enzymes include the SUMO-activating enzyme subunit 1/2 (SAE1/2) as E1, the Ubc9 as E2, and the protein inhibitor of activated STAT 1-4 (PIAS1-4), RanBP2, and the tripartite motif (TRIM) proteins as E3[30]. Of note, Ubc9 as the sole E2 enzyme forms a thioester bond with SUMO and directly interacts with substrates. Thus, Ubc9 has the capacity to add the SUMO modifiers to substrates, although E3 enzymes would facilitate this process[30]. Since Pin1 specifically elevates global SUMO1- but not SUMO2-modification in GSCs, isomerization of the Pin1 targets should affect the selective attachment of specific SUMO isoforms to substrates.

In this study, we investigate the upstream regulators and downstream effectors of Pin1 that is highly expressed in GSCs to induce hypersumoylation. We identify USP34 as a Pin1-interacting DUB that is responsible for the deubiquitination and stabilization of Pin1 proteins in GSCs. Of note, Plk1-mediated phosphorylation of Pin1 at S65 facilitates its binding to USP34, and the Plk1 inhibitor SBE13 reduces Pin1 protein expression. Furthermore, USP34 is preferentially expressed in GSCs, and disrupting USP34 impairs GSC maintenance and GBM growth. Interestingly, silencing USP34 severely inhibits Pin1-mediated SUMO1- but not SUMO2/3-modified sumoylation in GSCs. We further identify that the sole SUMO E2 enzyme Ubc9 is a Pin1 substrate in GSCs. Pin1 interacts with and isomerizes Ubc9 at the pS71-Pro motif. Noticeably, Pin1-mediated isomerization of Ubc9 selectively increases its affinity to SUMO1 but not SUMO2/3 modifier, resulting in the elevation of global SUMO1-modified sumoylation in GSCs. Moreover, we find that CDK1 promotes the phosphorylation of S71 at Ubc9 to enhance its interaction with Pin1. Finally, our pre-clinical study demonstrates that combination therapy with the Pin1 inhibitor sulfopin and the CDK1 inhibitor RO3306 most effectively suppresses SUMO1-modified sumoylation, inhibits GSC maintenance, and mitigates intracranial GBM growth. Our work not only discovers USP34 as the DUB regulating aberrant Pin1 expression at the post-translational level, but also reveals the deep involvement of Pin1 as a PPIase in protein sumoylation by directly isomerizing the sole SUMO E2 enzyme Ubc9. These discoveries provide multiple drug targets and therapeutic strategies for the treatment of GBMs and potentially other Pin1-driven cancers.

## Results

### Pin1 is deubiquitinated and stabilized by USP34 in GSCs

The oncogenic isomerase Pin1 is highly expressed in many cancers and plays key roles in multiple malignant aspects[4,15,31]. Whereas transcriptional and post-transcriptional regulations of Pin1 have

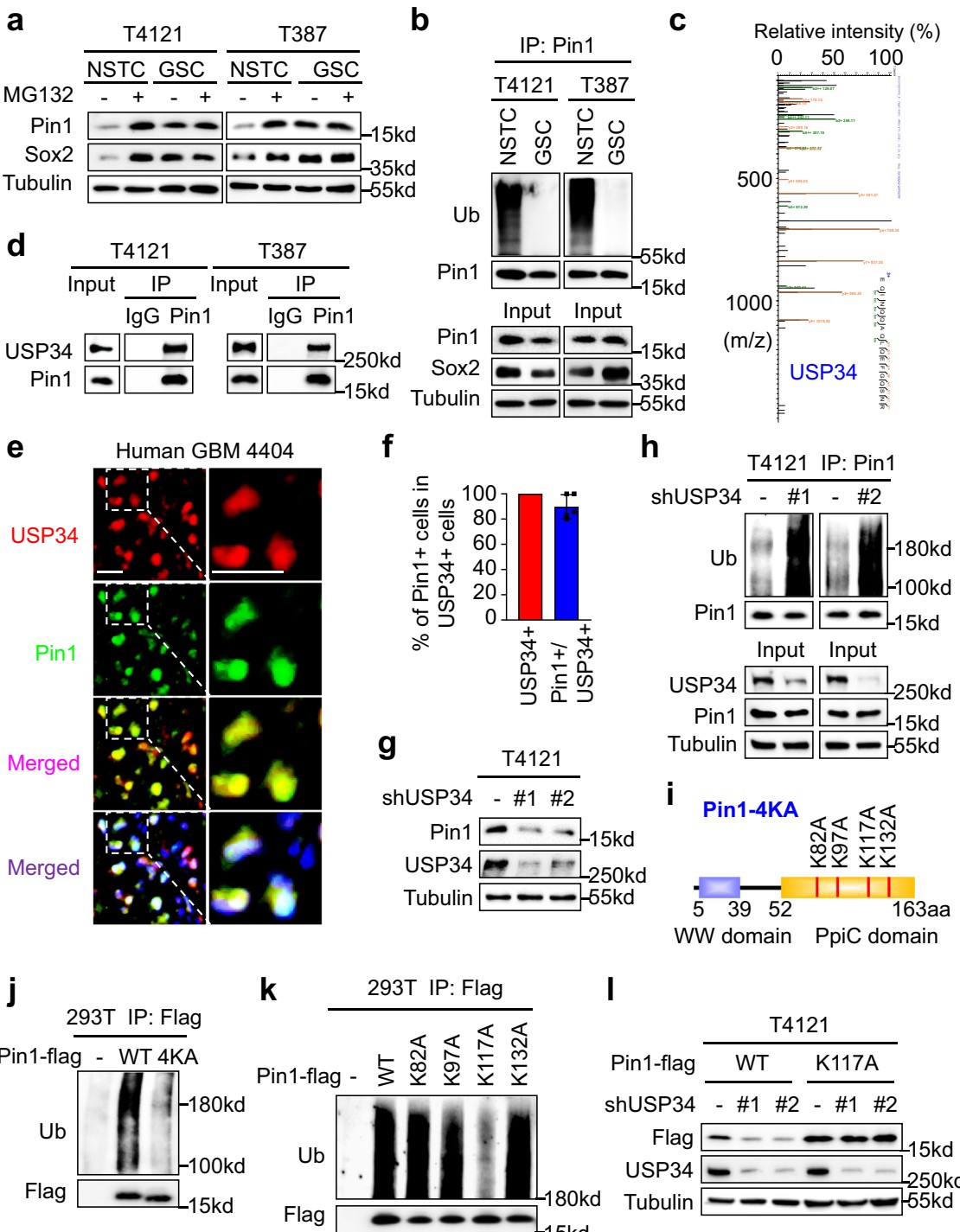

been recognized across different tumors, little is known about the regulation of Pin1 at the protein level[6,32]. Pin1 has been reported to be highly expressed in GSCs and functions to promote GSC maintenance and GBM growth, but the upstream regulatory mechanisms are elusive[4]. We sought to investigate the potential regulation of Pin1 at the protein level in GSCs. Treatment with the proteasome inhibitor MG132 showed negligible effects on Pin1 expression in GSCs but markedly restored Pin1 protein levels in the matched non-stem tumor cells (NSTCs) (Fig. 1a, Supplementary Fig. 1a). Meanwhile, Pin1 proteins had a strong poly-ubiquitination in NSTCs but not GSCs (Fig. 1b). These data strongly suggest a role of the ubiquitin-proteasome system (UPS) in regulating Pin1 expression. Protein ubiquitination is most often under the control of ubiquitin

E3 ligases and/or DUBs. Because Pin1 had much less ubiquitination in GSCs than NSTCs, we hypothesized that there may be a DUB responsible for Pin1 deubiquitination in GSCs. To search for the potential DUB of Pin1, Pin1-interacting proteins in GSCs were immunoprecipitated and analyzed by mass spectrometry. The results revealed that the DUB USP34 ranked first with the highest abundance among the precipitated Pin1-interacting proteins (Fig. 1c, Supplementary Fig. 1b). The interaction between USP34 and Pin1 in GSCs were validated by co-immunoprecipitation (Fig. 1d). To determine the correlation between USP34 and Pin1 in human GBM tissues, immunofluorescent staining of USP34 and Pin1 were performed on human primary GBM sections. The results showed that most USP34-positive tumor cells also had strong Pin1 staining

**Fig. 1 | Pin1 is deubiquitinated and stabilized by USP34 in GSCs. a** Immunoblot analysis of Pin1 in GSCs and NSTCs treated with MG132 (10 μM) or DMSO for 6 h. **b** Immunoblot analysis of the poly-ubiquitination of Pin1 in GSCs and NSTCs treated with MG132 (10 μM) for 6 h. **c** Identification of USP34 peptide in the Pin1-interacting proteins by mass spectrometry analysis. Pin1-interacting proteins were immuno-precipitated from GSCs (T4121) and digested with trypsin before LC-MS/MS analysis. A total of 50 unique peptides belonging to USP34 were detected. The spectra for the USP34 peptide EQINQQAQLQEFGQSNR is shown. **d** Co-immunoprecipitation to determine the interaction between Pin1 and USP34 in GSCs. **e, f** Representative images (**e**) and statistical quantification (**f**) of immuno-fluorescent analysis of USP34 (red) and Pin1 (green) in human primary GBMs. Scale bar, 20 μm. (*n* = 5 biological replicates; mean ± s.d.) **g** Immunoblot analysis of USP34 and Pin1 in T4121 GSCs expressing USP34-targeting shRNAs (shUSP34). **h** Immunoblot analysis of Pin1 polyubiquitination in T4121 GSCs expressing shUSP34. Cells were harvested 72 h post-lentiviral infection. MG132 (10 μM) was added 6 hours before harvest. **i** A schematic diagram of the Pin1-4KA mutant in which the four potentially ubiquitinated lysine residues were mutated to alanine. **j** Immunoblot analysis of the polyubiquitination of the wild type (WT) or the 4KA mutant Pin1 proteins ectopically expressed in 293 T cells. Flag-tagged Pin1 proteins were expressed in 293 T cells using polyethylenimine (PEI)-mediated transfection. Cells were harvested 48 h post-transfection, and MG132 (10 μM) was added 6 h before harvest. **k** Immunoblot analysis of polyubiquitination of the wild type Pin1 protein or the Pin1 proteins with indicated point-mutations in 293 T cells. **l** Immunoblot analysis of the protein levels of the wild type Pin1 and the Pin1-K117A mutant in T4121 GSCs expressing shUSP34. Ectopic flag-tagged Pin1 proteins were expressed in GSCs through lentiviral infection. Cells were further infected with shUSP34 lentiviruses and harvested 72 h post infection. The blotting experiments were repeated at least three times with biological replicates (**a**, **b**, **d**, **g**, **h**, **j**–**l**). Source data are provided as a Source Data file.

(Fig. 1e, f, Supplementary Fig. 1c). The above data indicate that USP34 as a Pin1-interacting DUB may positively regulate Pin1 expression in GSCs and GBMs.

Next, we investigated whether USP34 had the capacity to deubiquitinate and stabilize Pin1 proteins in GSCs. Disruption of the endogenous USP34 with lentiviral-mediated short hairpin RNAs (shRNAs) resulted in a remarkable reduction of Pin1 protein levels in GSCs, whereas Pin1 mRNA levels did not decrease (Fig. 1g, Supplementary Fig. 1d, e), indicating that USP34 regulated Pin1 expression at the protein level. Moreover, disruption of USP34 dramatically elevated poly-ubiquitination of Pin1 in GSCs (Fig. 1h, Supplementary Fig. 1f). Therefore, these data indicate that USP34 deubiquitinates and stabilizes Pin1 proteins. We then sought to identify the sites of poly-ubiquitination on Pin1 proteins. Four potential ubiquitination sites (K82, K97, K117, and K132) had been suggested in the Human Gene Database website (www.genecards.org), and accordingly we constructed a Pin1-4KA mutant by mutating all these four lysine residues to alanine (Fig. 1i). Whereas the ectopic wild-type Pin1 proteins showed strong ubiquitination in 293 T cells, the Pin1-4KA mutant showed almost no ubiquitination (Fig. 1j), suggesting that ubiquitination of Pin1 should occur at these four residues. To further determine the exact ubiquitination site, we constructed four single-residue Pin1 mutants. Ubiquitination assays were performed with the single-residue Pin1 mutants, and the reduction of ubiquitination was observed only on the Pin1-K117A mutant but not the other three mutants (Fig. 1k), indicating that the K117 residue should be the main ubiquitination site at Pin1. Moreover, when Pin1 proteins were ectopically expressed in GSCs, disruption of USP34 resulted in reduction of the wild-type Pin1 but not the K117A mutant (Fig. 1l, Supplementary Fig. 1g), suggesting that USP34 stabilized Pin1 by removing the poly-ubiquitination chain on the K117 residue. In summary, these data demonstrate that USP34 deubiquitinates and stabilizes Pin1 in GSCs.

## Phosphorylation of Pin1 by Plk1 enhances its interaction with UPS34

Given that USP34 interacts with and stabilizes Pin1, we sought to explore the mechanisms regulating the affinity of Pin1 to USP34. Protein interactions are frequently facilitated by phosphorylation, and Pin1 could be phosphorylated at multiple serine residues[6,32]. We initially examined the phosphor-serine (p-Ser) status of Pin1 in GSCs. Along with the higher Pin1 expression, Pin1 proteins showed stronger serine phosphorylation in GSCs relative to NSTCs (Fig. 2a), indicating a positive correlation between phosphorylation and stabilization of Pin1. We then explored the upstream kinase of Pin1. Because Plk1 had been reported to phosphorylate Pin1 and inhibit Pin1 ubiquitination[22], we first checked Plk1 expression in paired GSCs and NSTCs. Immunoblot analyses revealed a higher Plk1 expression in GSCs relative to NSTCs (Fig. 2b), which was reminiscent of the Pin1 expression. Furthermore, treatment with the Plk1 inhibitor SBE 13 remarkably reduced the Pin1

protein levels in GSCs (Fig. 2c). Moreover, immunoprecipitation analyses showed that SBE 13 treatment markedly inhibited serine phosphorylation of Pin1 (Fig. 2d), even when Pin1 protein levels were restored with MG132 treatment (Fig. 2d). These data suggest that Plk1 promotes serine phosphorylation of Pin1 to stabilize Pin1 proteins in GSCs.

We then investigated whether Plk1-mediated phosphorylation of Pin1 affected the interaction between Pin1 and USP34. Co-immunoprecipitation analyses showed that inhibition of Plk1 by SBE 13 attenuated the interaction between Pin1 and USP34 (Fig. 2e). Moreover, SBE 13 treatment elevated poly-ubiquitination of Pin1 proteins, which was likely resulted from the dissociation of the DUB USP34 from Pin1 proteins (Fig. 2f). Previous studies had reported that Plk1 phosphorylates the S65 residue on Pin1 protein. In order to determine whether phosphorylation of S65 in Pin1 promoted the binding of USP34, we constructed a Pin1 S65A mutant by substitution of serine with alanine. The wild type (WT) Pin1 or the Pin1-S65A mutant were introduced into 293 T cells, and the endogenous Pin1 was disrupted with a shRNA targeting the 3'UTR of Pin1 mRNA. Affinities of the ectopic Pin1 proteins to endogenous USP34 were determined with co-immunoprecipitation analysis. The results showed that Pin1-S65A relative to the wildtype Pin1 had a much lower affinity to USP34 (Fig. 2g). In line with the decreased interaction with USP34, the Pin1-S65A mutant had an elevated poly-ubiquitination relative to the wild type Pin1 (Fig. 2h). Taken together, these data indicate that Plk1-mediated phosphorylation of S65 in Pin1 promotes the binding of USP34 and the consequent deubiquitination of Pin1 in GSCs.

## USP34 is preferentially expressed in GSCs to promote GSC maintenance and GBM tumor growth

Pin1 as a pivotal oncogene promotes malignant progression of several cancers[4,15,31]. Whereas the high expression of Pin1 is indispensable for GSC maintenance[4], the roles of USP34 in GSCs and GBMs remain poorly understood. Given that USP34 stabilized Pin1 proteins, we proposed that USP34 may have an important role in GSC maintenance. We initially determined the expression of USP34 in GSCs. Immunoblot analyses detected much higher levels of USP34 in GSCs relative to NSTCs, which was consistent with the preferential expression of Pin1 in GSCs (Fig. 3a). To further investigate USP34 expression in human GBM tissues, we performed immunofluorescent staining on frozen human GBM sections. The results showed that most GSCs marked by SOX2 or OLIG2 had USP34 expression, whereas NSTCs had rare USP34 staining (Fig. 3b, Supplementary Fig. 2a–c). Meanwhile, the majority of USP34 positive cells were positively stained with SOX2 and OLIG2 (Fig. 3b, c, Supplementary Fig. 2a, b). These data indicate a preferential expression of USP34 in GSCs rather than NSTCs, suggesting a potential role of USP34 in GSC maintenance.

Next, we disrupted USP34 in GSCs to interrogate its role in GSC maintenance (Fig. 3d, Supplementary Fig. 2d). Tumorsphere formation

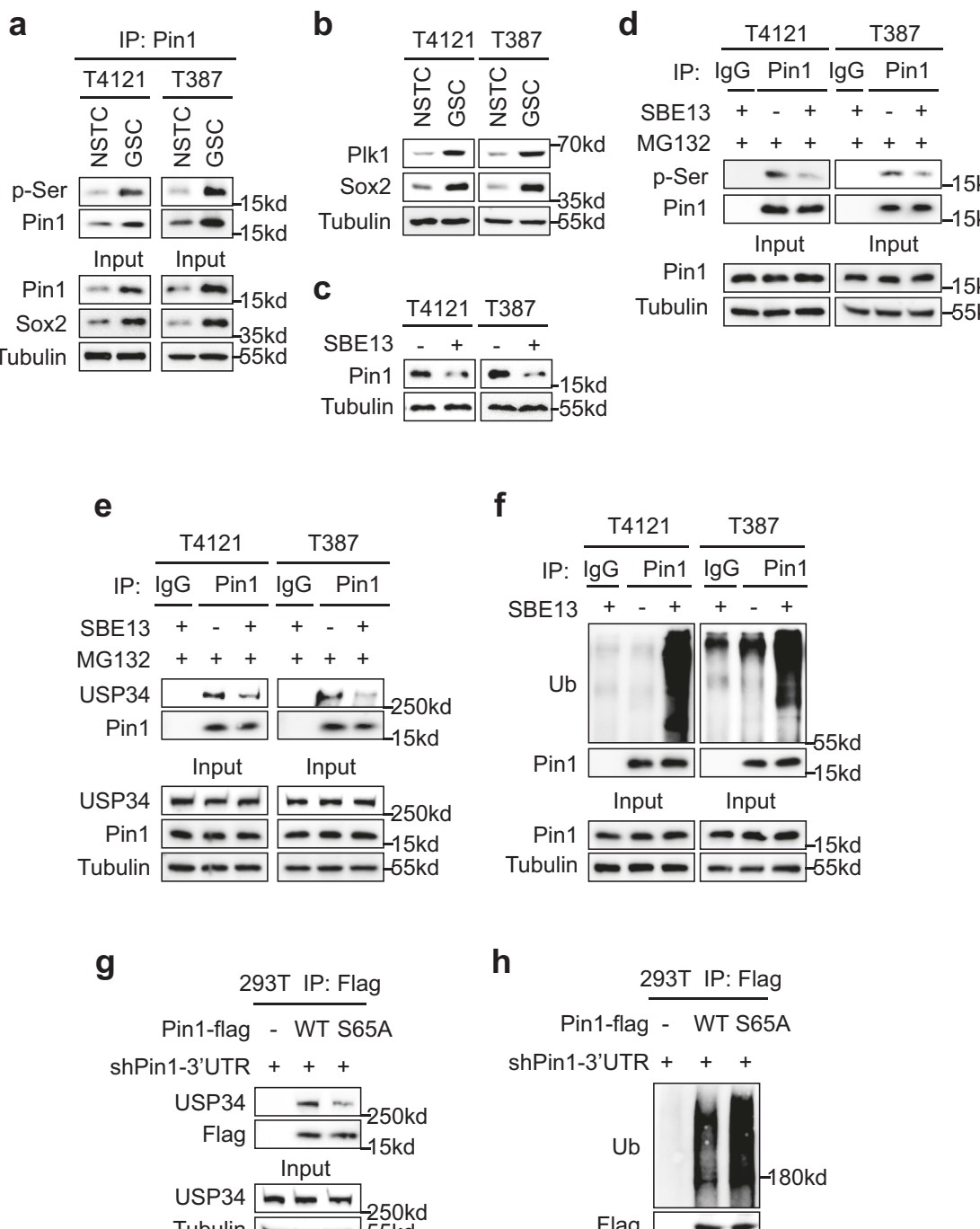

**Fig. 2 | Phosphorylation of Pin1 by Plk1 enhances its interaction with UPS34.**
**a** Immunoblot analysis of the serine phosphorylation (pSer) status of Pin1 in GSCs and NSTCs. **b** Immunoblot analysis of Plk1 protein levels in GSCs and NSTCs. **c** Immunoblot analysis of Pin1 protein levels in GSCs treated with the Plk1 inhibitor SBE 13 HCl (20 μM) or DMSO for 24 h. **d** Immunoblot analysis of the serine phosphorylation (pSer) status of Pin1 in GSCs after inhibition of Plk1. Cells were treated with SBE 13 HCl (20 μM) or DMSO for 12 h. MG132 (10 μM) was added 6 h before harvest to obtain enough Pin1 proteins after Plk1 inhibition. **e** Co-immunoprecipitation to determine the interaction between Pin1 and USP34 in GSCs after inhibition of Plk1. Cells were treated with SBE 13 HCl (20 μM) or DMSO for 9 h and harvested. MG132 (10 μM) was added 6 h before harvest to obtain enough Pin1 proteins after Plk1 inhibition. **f** Immunoblot analysis of the poly-ubiquitination status of Pin1 in GSCs after inhibition of Plk1. Cells were treated with SBE 13 HCl

(20 μM) or DMSO for 9 h and harvested. MG132 (10 μM) was added 6 h before harvest. **g** Co-immunoprecipitation to determine the interaction between USP34 and the wild type (WT) Pin1 or the Pin1-S65A mutant. Serine 65 (S65) at Pin1 was mutated to alanine to construct the Pin1-S65A mutant. Flag-tagged Pin1-WT or Pin1-S65A was overexpressed in 293 T cells. Expression of the endogenous Pin1 was disrupted in 293 T cells with an shRNA targeting the 3'-untranslated region (3'-UTR) of Pin1 (shPin1-3'UTR). Overexpression of ectopic Pin1 and disruption of endogenous Pin1 were achieved by PEI-mediated transfection. **h** Immunoblot analysis of the poly-ubiquitination status of Pin1-WT or Pin1-S65A proteins in 293 T cells. Flag-tagged Pin1-WT or Pin1-S65A was introduced into 293 T cells along with shPin1-3'UTR. Cells were treated with MG132 for 6 hours before harvest. The blotting experiments were repeated at least three times with biological replicates (**a–h**). Source data are provided as a Source Data file.

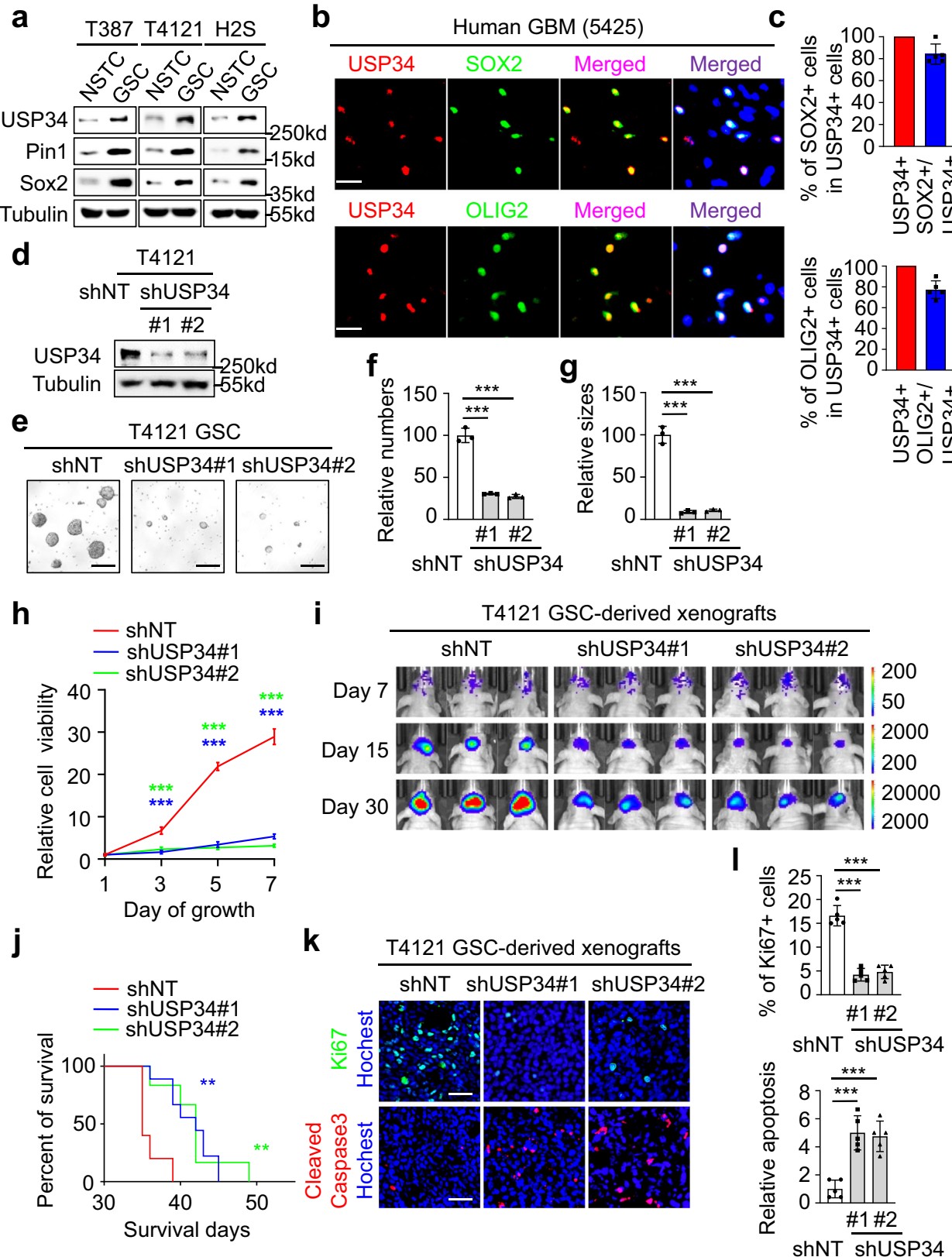

assays demonstrated that disruption of USP34 markedly inhibited GSC sphere formation, which was represented by a dramatic decrease in both sphere numbers and sizes (Fig. 3e–g, Supplementary Fig. 2e–g). Furthermore, cell viability assays showed that disruption of USP34 had a severe impact on GSC proliferation (Fig. 3h, Supplementary Fig. 2h). These phenomena demonstrate that USP34 functions to promote GSC

maintenance. We then explored the role of USP34 in GBM tumor growth in mouse brains. GSCs expressing USP34 shRNAs or non-targeting shRNA and firefly luciferase were transplanted into immunocompromised mice through intracranial injection. Bioluminescent imaging of orthotopic tumors in vivo showed a markedly delayed growth of xenografts derived from GSCs expressing USP34 shRNA

**Fig. 3 | USP34 is preferentially expressed in GSCs to promote GSC maintenance and GBM tumor growth. a** Immunoblot analysis of USP34 in GSCs and NSTCs. **b**, **c** Representative images (**b**) and statistical quantification (**c**) of immuno-fluorescent analysis of USP34 (red) and SOX2 or OLIG2 (green) in human primary GBMs. Scale bar, 20 μm. ($n = 5$ biological replicates; mean ± s.d.) **d** Immunoblot analysis of USP34 in T4121 GSCs expressing shUSP34. **e** Representative images of tumorsphere formation of T4121 GSCs expressing shUSP34 or shNT. Forty-eight hours after lentiviral infection, 2000 cells were planted in each well of 96-well plates and cultured for 5 days. Scale bar, 50 μm. **f**, **g** Statistical quantifications of tumorsphere numbers (**f**) and sizes (**g**) of T4121 GSCs expressing shUSP34 or shNT. Three random 20× fields were used for calculation. ($n = 3$ biological replicates; ***$P < 0.001$; mean ± s.d.; two tailed unpaired $t$-test). **h** Cell proliferation of T4121 GSCs expressing shUSP34 or shNT. Forty-eight hours after lentiviral infection, 2000 cells were planted in each well of 96-well plates and cell viability was determined at the indicated time points. ($n = 4$ biologically independent experiments; ***$P < 0.001$; mean ± s.d.; two-way ANOVA). **i** In vivo bioluminescent imaging ana-lysis of intracranial growth of GBM xenografts derived from T4121 GSCs expressing shUSP34 or shNT. Five-thousand GSCs were injected into mouse brains and representative images at the indicated days post-transplantation were shown. **j** Kaplan-Meier survival curves of tumor-bearing mice in (**i**). (shNT, $n = 5$ mice; shUSP34#1, $n = 7$ mice; shUSP34#2, $n = 6$ mice; shUSP34#1 vs. shNT, $P = 0.0018$; shUSP34#2 vs. shNT, $P = 0.0038$; **$P < 0.01$; two-tailed log-rank test.) **k**, **l** Immunofluorescent analysis (**k**) and statistical quantifications (**l**) of Ki67 (green) and cleaved caspase-3 (red) in GBM xenografts. Scale bar, 40 μm ($n = 5$ tumors for each group; ***$P < 0.001$; mean ± s.d.; two-tailed unpaired $t$-test). The blotting experiments were repeated at least three times with biological replicates (**a**, **d**). Source data are provided as a Source Data file.

relative to those expressing non-targeting shRNA (Fig. 3i, Supplementary Fig. 2i). Consistently, disruption of USP34 extended the survival of mice bearing the GSC-derived tumors (Fig. 3j, Supplementary Fig. 2j). Immunofluorescent analyses of the resultant xenografts with the apoptotic marker cleaved caspase-3 and the cell proliferation marker Ki67 showed that silencing USP34 dramatically elevated apoptosis but inhibited proliferation of tumor cells (Fig. 3k, l, Supplementary Fig. 2k, l). Collectively, these data demonstrate that USP34 promotes GSC maintenance and GBM tumor growth.

## USP34 promotes Pin1-mediated SUMO1-modified protein sumoylation in GSCs

Disruption of USP34 showed a severe impact on GSC maintenance and GBM growth, which was reminiscent of the outcome of Pin1 silencing[4]. The critical functions of Pin1 in GSCs had been ascribed to its capacity to elevate global SUMO1-modified protein sumoylation[4]. We therefore investigated the role of USP34 as a Pin1 upstream regulator in protein sumoylation in GSCs. As expected, immunoblot analyses showed that disruption of USP34 dramatically reduced SUMO1- but not SUMO2/3-sumoylation in GSCs (Fig. 4a, Supplementary Fig. 3a), which was similar to the effect of Pin1 silencing[4]. Likewise, immunofluorescent analyses of the GSC-derived xenografts showed that whereas a substantial portion of cells in the control tumors had intensive SUMO1 staining, disruption of USP34 resulted in a remarkable reduction of SUMO1 signals in tumor tissues (Fig. 4b, c, Supplementary Fig. 3b, c). Meanwhile, negligible change of SUMO2/3 staining was observed in tumor tissues after USP34 silencing (Fig. 4b, c, Supplementary Fig. 3b, c). Furthermore, immunofluorescent analyses of human GBM samples demonstrated that the majority (>80%) of tumor cells with SUMO1 staining had strong USP34 expression (Fig. 4d, e, Supplementary Fig. 3d, e). However, less than 50% of tumor cells with SUMO2/3 staining had USP34 signals (Fig. 4d, e, Supplementary Fig. 3d, e). Importantly, the majority of SOX2+ GSCs were positively stained for both USP34 and SUMO1 (Supplementary Fig. 3f, g). Of note, whereas very few SOX2- NSTCs had high SUMO1-sumoylation, these NSTCs hardly had USP34 signals in immunofluorescent staining (Supplementary Fig. 3f). These data strongly suggest that USP34-should be an upstream regulator of SUMO1- but not SUMO2/3-sumoylation in GSCs. We proposed that USP34 promoted protein SUMO1-sumoylation via stabilization of Pin1. Therefore, Pin1 should have the ability to restore the reduced protein sumoylation resulted from USP34 silencing. The ubiquitination-deficient Pin1-K117A mutant that was stable even after disruption of USP34 (Fig. 1l) was applied to test the above hypothesis. Indeed, immunoblot analyses showed that although disruption of USP34 almost abrogated SUMO1-sumoylation, ectopic expression of Pin1-K117 largely restored protein SUMO1-sumoylation in GSCs (Fig. 4f, Supplementary Fig. 3h). Taken together, the above data demonstrate that USP34 upregulates SUMO1-modified protein sumoylation via Pin1 in GSCs.

## Pin1 isomerizes Ubc9 to facilitate formation of the Ubc9-SUMO1 thioester

Because the stabilization of Pin1 by USP34 promoted SUMO1-modified protein sumoylation in GSCs, we sought to explore the Pin1 isomerase substrates involved in hypersumoylation. Pin1 is a highly conserved enzyme that specifically isomerizes the phosphorylated Ser/Thr-Pro motif in substrate proteins. We proposed that Pin1 may regulate protein conformation and function of the key components of sumoylation machinery that controls global sumoylation. Ubc9 is the sole SUMO E2 conjugation enzyme that not only transfers but also ligates the activated SUMO modifiers to the substrate proteins[33]. Interestingly, Ubc9 contains one Ser/Thr-Pro motif composed of serine 71 (S71) and proline 72 (P72), which may be subject to Pin1-catalyzed isomerization (Fig. 5a). To determine whether Ubc9 could be a substrate of Pin1, we initially checked the association between Ubc9 and Pin1 in GSCs. Co-immunoprecipitation analyses detected a strong interaction between Ubc9 and Pin1 (Fig. 5b, Supplementary Fig. 4a). We then explored whether Ubc9 could be isomerized by Pin1 through in vitro isomerization assays[12,14,34,35]. Spectrophotometric method was applied to determine the Pin1-catalyzed isomerization of the chromogenic Ubc9 oligopeptides containing phosphorylated or nonphosphorylated S71. The increase of the 4-nitroanilide products released from the Ubc9 peptides reflected the configuration change of the S71-P72 bond. Incubation with the purified Pin1 proteins markedly elevated the 4-nitroanilide production from the Ubc9-pS71 peptide but not the nonphosphorylated counterpart (Fig. 5c, Supplementary Fig. 4b, c), indicating that the Ubc9 pS71-Pro motif was isomerized by Pin1. Conformational change may affect protein stability and thereby Pin1 may regulate Ubc9 expression at the protein level. However, although immunoblot analyses revealed a higher protein expression of Ubc9 in GSCs relative to NSTCs (Supplementary Fig. 4d), disruption of Pin1 in GSCs did not affect Ubc9 levels (Supplementary Fig. 4e), indicating that Pin1-mediated isomerization did not regulate Ubc9 protein expression. Collectively, these data show that Pin1 interacts with and isomerizes Ubc9 in GSCs.

Next, we sought to investigate the connection between Pin1-catalyzed isomerization of Ubc9 and SUMO1-modified protein sumoylation. In the process of protein sumoylation, a thioester bond is formed between the SUMO modifier and the catalytic cysteine residue of Ubc9, which is a prerequisite for the Ubc9-mediated ligation of SUMO to substrates[36]. We therefore explored whether Pin1-catalyzed isomerization may affect the Ubc9 thioester formation with SUMO1. The covalent attachment of the SUMO1 modifier to Ubc9 through the thioester bond would generate a Ubc9-SUMO1 conjugate that migrates with a molecular weight of ~37 kDa[33,37]. However, the reducing reagents dithiothreitol (DTT) and β-mercaptoethanol (β-ME) could disrupt the thioester bond and release the SUMO1 modifier[33,38,39]. When ectopic myc-tagged Ubc9 in 293 T cells was precipitated with anti-myc antibodies, both anti-myc and anti-SUMO1 antibodies detected a clear

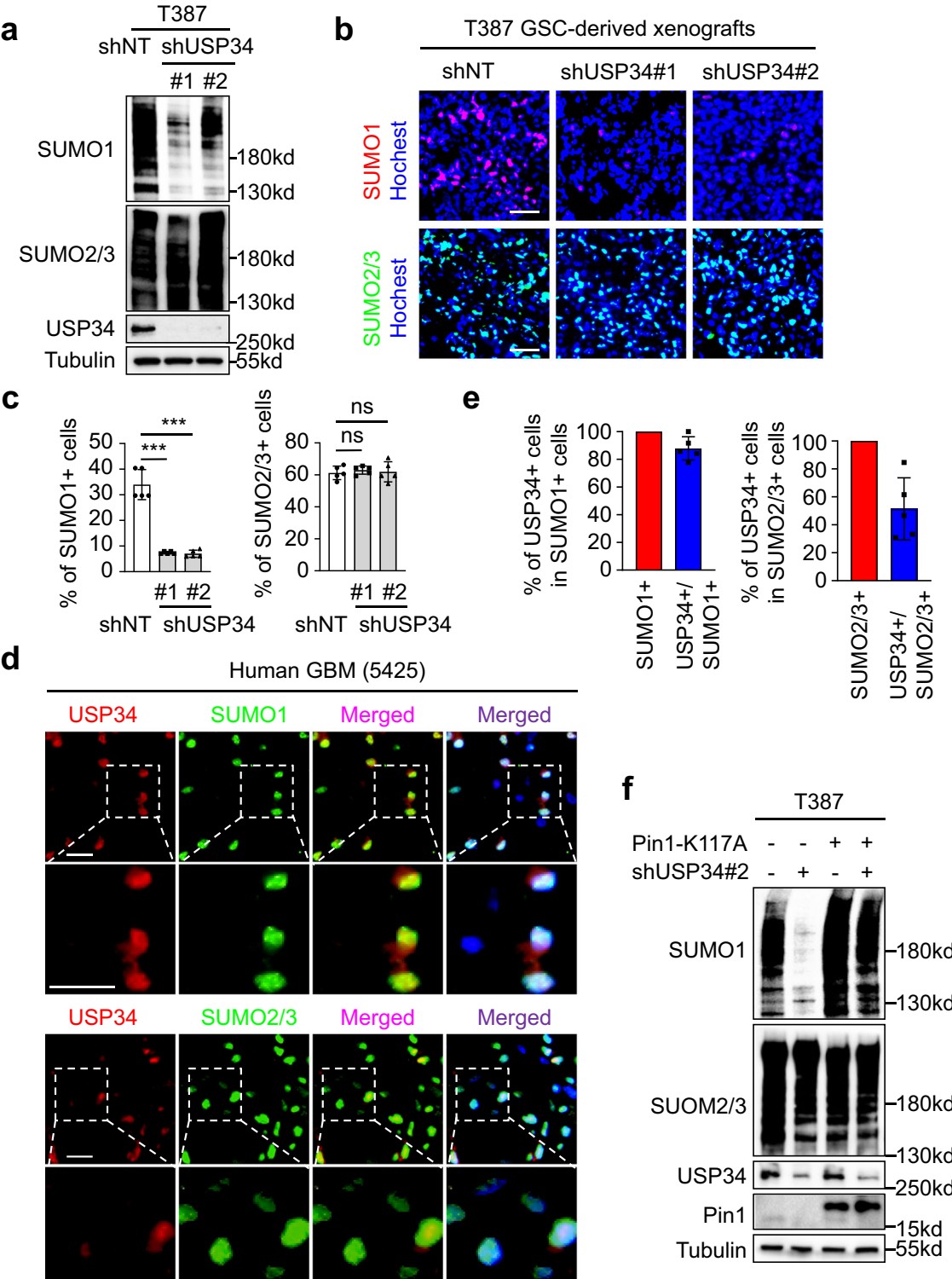

**Fig. 4 | USP34 promotes Pin1-mediated SUMO1-modified protein sumoylation in GSCs. a** Immunoblot analysis of global protein sumoylation status in T387 GSCs expressing shUSP34 or shNT. **b, c** Immunofluorescent analysis (**b**) and statistical quantifications (**c**) of protein sumoylation with SUMO1 (red) and SUMO2/3 (green) modification in xenografts derived from T387 GSCs expressing shUSP34 or shNT. Scale bar, 40 μm (*n* = 5 tumors for each group; ***P < 0.001; mean ± s.d.; two-tailed unpaired *t*-test). **d, e** Representative images (**d**) and statistical quantification (**e**) of immunofluorescent analyses of USP34 (red) and SUMO1 or SUMO2/3 (green) in human primary GBMs. Frozen sections of human GBMs were immunostained with antibodies against USP34 and SUMO1 or SUMO2/3, and counterstained with Hoechst to show nuclei (blue). The percentage of USP34+ cells in the SUMO1+ or SUMO2/3+ population was quantified. Scale bar, 20 μm. (n = 5 biological replicates;

mean ± s.d.) **f** Immunoblot analysis of global protein sumoylation status in T387 GSCs expressing shUSP34 and the ectopic Pin1-K117A mutant. GSCs were transduced with the ubiquitination-deficient Pin1-K117A mutant through lentiviral infection. The cells were further infected with shUSP34 lentivirus and the cell lysates were subjected to immunoblot analysis with indicated antibodies. The Pin1-K117A protein was stable even though the endogenous USP34 was reduced by shUSP34. Meanwhile, whereas disruption of USP34 reduced SUMO1- but not SUMO2/3-modified sumoylation, ectopic expression of Pin1-K117 restored SUMO1-modified sumoylation in GSCs. The blotting experiments were repeated at least three times with biological replicates (**a**, **f**). Source data are provided as a Source Data file.

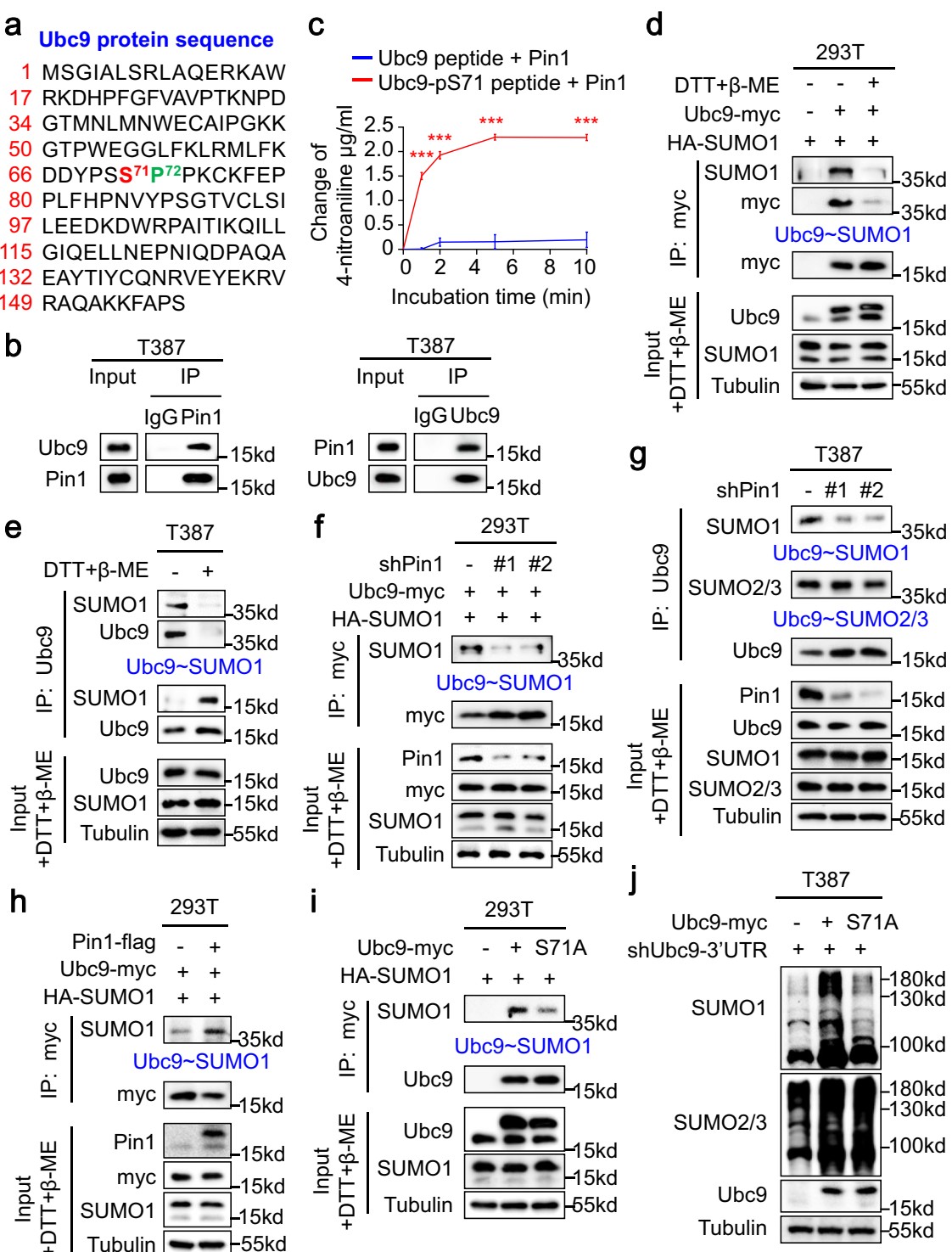

band at ~37 kDa (Fig. 5d). This band was proven to be the Ubc9-SUMO1 conjugate, because addition of DTT and β-ME almost eliminated the band (Fig. 5d). In the meantime, the band of the unconjugated Ubc9 at ~18 kDa showed a slight increase after the addition of DTT and β-ME (Fig. 5d), reflecting the disruption of the Ubc9-SUMO1 conjugate and the resultant accumulation of free Ubc9. Likewise, immunoprecipitation with the anti-Ubc9 antibodies pulled down the Ubc9-SUMO1 conjugate composed of endogenous Ubc9 and SUMO1 at a molecular weight of ~37 kDa in GSCs (Fig. 5e). Addition of DTT and β-ME markedly reduced the amount of the Ubc9-SUMO1 conjugate, resulting in the release and accumulation of free SUMO1 modifiers along with a mild

increase of unconjugated Ubc9 (Fig. 5e). We then investigated the role of Pin1 in the formation of the Ubc9-SUMO1 conjugate. Surprisingly, disruption of Pin1 in 293 T cells resulted in a dramatic decrease of the Ubc9-SUMO1 conjugate (Fig. 5f), indicating the inhibition of the Ubc9-SUMO1 thioester formation. Of note, Pin1 silencing did not affect the formation of the Ubc9-SUMO2/3 conjugate composed of ectopic Ubc9 and endogenous SUMO2/3 in 293 T cells, which was detected by the anti-SUMO2/3 antibodies at a molecular weight of ~37 kDa (Supplementary Fig. 4f). Similarly, disruption of the endogenous Pin1 severely impacted the formation of the Ubc9-SUMO1 conjugate in GSCs (Fig. 5g, Supplementary Fig. 4g), whereas the Ubc9-SUMO2/3

**Fig. 5 | Pin1 isomerizes Ubc9 to facilitate formation of the Ubc9-SUMO1 thioester. a** A diagram of the Ubc9 protein sequence depicting the potential phosphorylated serine-proline (pS-Pro) motif composed of the serine 71 and proline 72 residues. **b** Co-immunoprecipitation to determine the interaction between Pin1 and Ubc9 in T387 GSCs. **c** Cis-trans isomerization assay to determine the Pin1-catalyzed isomerization of the phosphorylated Ubc9. The phosphorylated Ubc9 oligopeptides (Ubc9-pS71) or the control peptides were incubated with the purified flag-tagged Pin1. The 4-nitroaniline released from the Ubc9 peptides was measured continually, and its change reflected the difference in the isomerization of prolyl bonds. Data represent three independent experiments (***$P < 0.001$; means ± s.d. two-way ANOVA). **d, e** Immunoblot analyses of the Ubc9-SUMO1 thioester (Ubc9 ˜ SUMO1) formation in 293 T cells expressing ectopic Ubc9 and SUMO1 (**d**) and in T387 GSCs (**e**). The immunoprecipitated Ubc9-SUMO1 conjugate located at ~37 kDa. Addition of DTT (200 mM) and β-ME (5%) into the loading buffer broke the Ubc9-SUMO1 thioester, leading to the disappearance of the Ubc9-SUMO1 conjugate and the increase of free SUMO proteins at ~15 kDa. For a fair comparison for input, DTT and β-ME were added before loading to disrupt any potential thioesters. **f** Immunoblot analysis of the Ubc9-SUMO1 thioester (Ubc9 ˜ SUMO1) formation in 293 T cells after disruption of endogenous Pin1. The immunoprecipitated Ubc9-SUMO1 conjugate was detected by anti-SUMO1 antibodies. **g** Immunoblot analysis of the Ubc9-SUMO1 thioester (Ubc9 ˜ SUMO1) and Ubc9-SUMO2/3 thioester (Ubc9 ˜ SUMO2/3) formation in T387 GSCs after disruption of Pin1. **h** Immunoblot analysis of the Ubc9-SUMO1 thioester (Ubc9 ˜ SUMO1) formation in 293 T cells overexpressing ectopic Pin1. **i** Immunoblot analysis of the Ubc9-SUMO1 thioester (Ubc9 ˜ SUMO1) formation in 293 T cells expressing ectopic wild type Ubc9 or the Ubc9-S71A mutant Ubc9 and SUMO1. **j** Immunoblot analysis of global protein sumoylation status in T387 GSCs expressing ectopic Ubc9. GSCs were transduced with the wild type Ubc9 or the Ubc9-S71A mutant through lentiviral infection followed by infection with lentiviruses carrying an shRNA targeting 3′-UTR of Ubc9 (shUbc9-3′UTR). The blotting experiments were repeated at least three times with biological replicates (**b, d–j**). Source data are provided as a Source Data file.

conjugate remained intact (Fig. 5g, Supplementary Fig. 4g). Moreover, overexpression of Pin1 in 293 T cells enhanced the formation of the Ubc9-SUMO1 but not the Ubc9-SUMO2/3 conjugate (Fig. 5h, Supplementary Fig. 4h). The above data strongly suggest that Pin1 promotes the formation of Ubc9 thioester with SUMO1 but not SUMO2/3. Furthermore, we addressed whether the mutation of the pS71-Pro motif in Ubc9 to inhibit Pin1-catalyzed isomerization would affect the conjugation of SUMO1 and SUMO2/3 modifiers to Ubc9. As expected, while strong signals of the Ubc9-SUMO1 conjugate was detected between the ectopic wild type Ubc9 and SUMO1 in 293 T cells, formation of the Ubc9-SUMO1 conjugate was much weaker between the ectopic Ubc9-S71A mutant and SUMO1 (Fig. 5i). Furthermore, overexpression of Pin1 markedly elevated the conjugation of SUMO1 to the wild type Ubc9 but not the Ubc9-S71A mutant, whereas the high levels of Ubc9-SUMO2/3 conjugates were not affected by neither Pin1 overexpression nor mutation of the pS71-Pro motif (Supplementary Fig. 4i), indicating that Pin1-catalyzed isomerization of Ubc9 increases its affinity to SUMO1 without antagonizing SUMO2/3 conjugation in general. Finally, when disruption of the endogenous Ubc9 severely reduced global sumoylation, ectopic expression of the wild type but not the Ubc9-S71A mutant restored SUMO1-modified protein sumoylation in GSCs (Fig. 5j, Supplementary Fig. 4j). However, the wild type Ubc9 and the Ubc9-S71A mutant demonstrated similar ability to restore SUMO2/3-modified protein sumoylation (Fig. 5j, Supplementary Fig. 4j). In summary, these data show that Pin1-catalyzed isomerization of Ubc9 enhances the Ubc9 thioester formation with SUMO1 but not SUMO2/3 modifier, which in turn leads to elevated global SUMO1-modified protein sumoylation in GSCs.

### CDK1 phosphorylates Ubc9 to facilitate its interaction with Pin1

Phosophorylation at the Ser/Thr-Pro motif in the substrate protein is required for Pin1-catalyzed isomerization, therefore the S71 residue in Ubc9 as a Pin1 substrate is supposed to be phosphorylated. Immunoprecipitation analyses showed that whereas the wild type Ubc9 had strong serine phosphorylation, mutation of the S71 residue almost eliminated the phosphorylation of Ubc9 (Fig. 6a), suggesting that S71 was the major phosphorylation site in Ubc9 in GSCs. Interestingly, previous studies had demonstrated that phosphorylation of S71 in Ubc9 by the CDK1 kinase promoted SUMO1-modified protein sumoylation[38]. We therefore investigated the regulatory role of CDK1 in the interaction between Pin1 and Ubc9, which was prerequisite for Pin1-catalyzed Ubc9 isomerization and the consequent SUMO1-modified hypersumoylation in GSCs. Immunoblot analyses detected a higher expression of CDK1 in GSCs relative to NSTCs, which was consistent with the higher global SUMO1-modified sumoylation in GSCs (Fig. 6b). Moreover, inhibition of CDK1 with the specific inhibitor RO3306 almost abolished the serine phosphorylation of Ubc9 just like the mutation of S71 residue (Fig. 6a, c), indicating the CDK1 was responsible for phosphorylation of S71 in Ubc9. Importantly, co-immunoprecipitation analyses showed that inhibition of CDK1 by RO3306 markedly attenuated the interaction between Ubc9 and Pin1 in GSCs (Fig. 6d), suggesting that CDK1 facilitated the binding of Pin1 to Ubc9. To further clarify if the phosphorylated S71 residue in Ubc9 is the binding site for Pin1, we deployed the Ubc9-S71A mutant and determined its affinity to Pin1. Co-immunoprecipitation analyses showed that mutation of S71 in Ubc9 remarkably attenuated the interaction between Ubc9 and Pin1 (Fig. 6e). Meanwhile, inhibition of CDK1 did not further reduce the interaction between the Ubc9-S71A mutant and Pin1 (Fig. 6f), suggesting that S71 phosphorylated by CDK1 was the dominant binding site for Pin1. Finally, inhibition of CDK1 by RO3306 markedly attenuated global SUMO1- but not SUMO2/3-modified sumoylation in GSCs (Fig. 6g), reflecting the inhibition of Pin1-catalyzed Ubc9 isomerization and the subsequent sumoylation. Taken together, these data demonstrate that the S71 residue in Ubc9 is phosphorylated by CDK1 to facilitate the binding of Pin1, which is critical for the Ubc9 isomerization and the SUMO1-modified hypersumoylation in GSCs.

### Simultaneous inhibition of Pin1 and CDK1 suppresses SUMO1-modified sumoylation in GSCs and mitigates GBM growth

We have shown that Pin1 is deubiquitinated and stabilized by USP34 in GSCs, resulting in the Pin1-catalyzed Ubc9 isomerization and the consequent SUMO1-modified hypersumoylation that are critical for GSC maintenance. Meanwhile, the CDK1 kinase is responsible for Ubc9 phosphorylation and the recognition of Ubc9 by Pin1. Therefore, both the isomerase activity of Pin1 and the kinase activity of CDK1 may be key targets for inhibition of SUMO1-modified protein sumoylation, and simultaneous inhibition of Pin1 and CDK1 should strongly benefit GBM treatment. To test this hypothesis, GSCs were treated with the Pin1 inhibitor sulfopin and the CDK1 inhibitor RO3306. Either sulfopin or RO3306 treatment alone severely impacted GSC cell viability, but sulfopin plus RO3306 treatment almost eliminated GSCs (Fig. 7a, Supplementary Fig. 5a). Immunoblot analyses showed that while both sulfopin and RO3306 reduced SUMO1-modified sumoylation in GSCs (Fig. 7b, Supplementary Fig. 5b), combined treatment with sulfopin plus RO3306 resulted in the most impressive decrease of SUMO1-sumoylation (Fig. 7b, Supplementary Fig. 5b). Meanwhile, SUMO2/3-sumoylation was not affected by sulfopin or RO3306 (Fig. 7b, Supplementary Fig. 5b). These data suggest that simultaneous inhibition of Pin1 and CDK1 suppresses SUMO1-modified sumoylation and impairs GSC maintenance. We then applied sulfopin and RO3306 to treat GSC-derived orthotopic GBMs. Either sulfopin or RO3306 treatment alone delayed tumor growth, as indicated by the decreased intracranial bioluminescent signals from drug-treated mice relative to the control group (Fig. 7c). However, combined treatment with sulfopin plus RO3306 most effectively inhibited tumor growth (Fig. 7c).

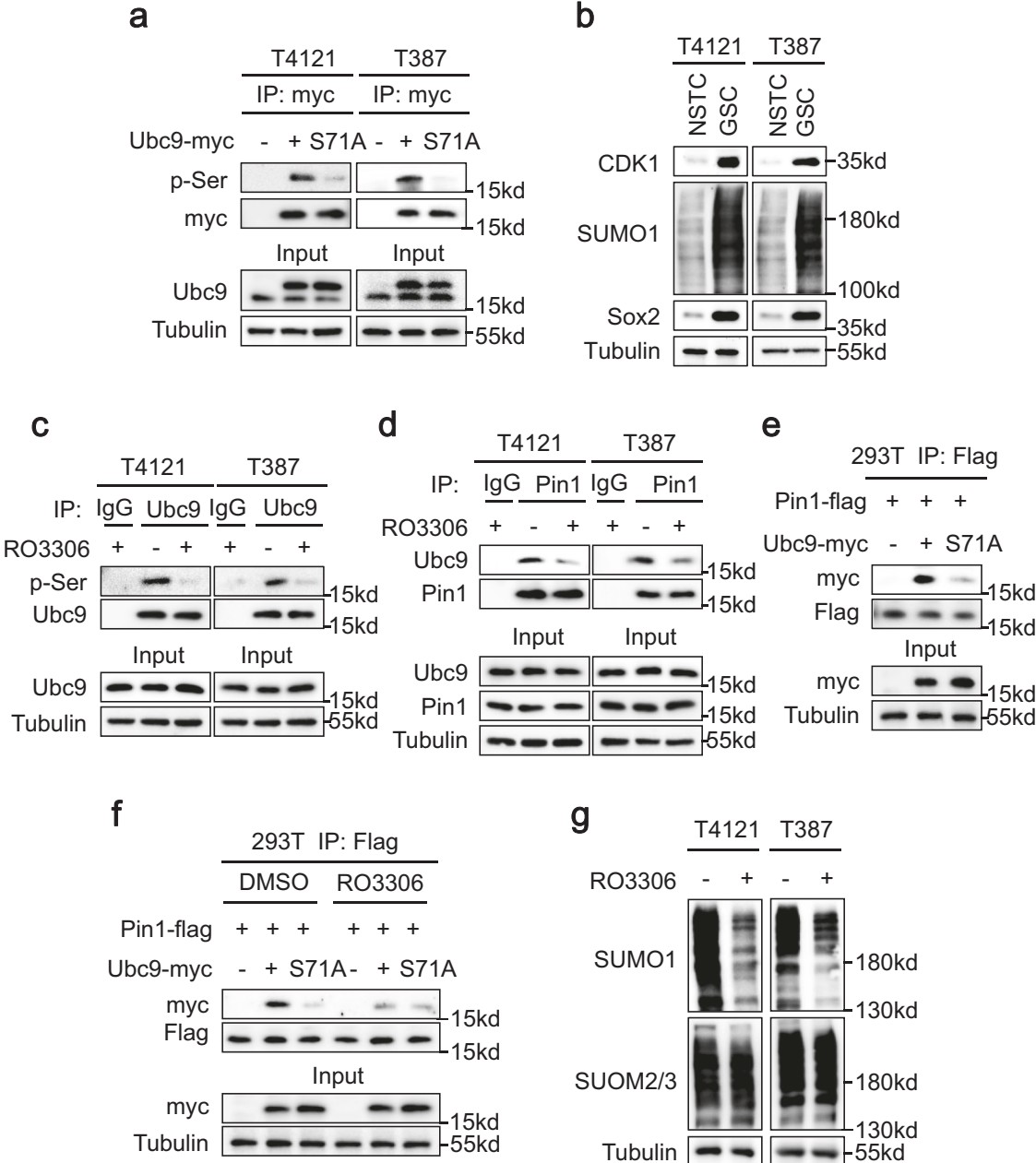

**Fig. 6 | CDK1 phosphorylates Ubc9 to facilitate its interaction with Pin1.**
**a** Immunoblot analysis of the serine phosphorylation (pSer) status of the wild type Ubc9 and the Ubc9-S71A mutant in T4121 and T387 GSCs. Immunoprecipitated Ubc9 proteins were subjected to immunoblot with anti-pan-pSer antibodies.
**b** Immunoblot analysis of CDK1 levels and global SUMO1-modified sumoylation status in GSCs and NSTCs. **c** Immunoblot analysis of the serine phosphorylation (pSer) status of Ubc9 in GSCs treated with the CDK1 inhibitor RO3306 (20 µM) or DMSO for 12 h. **d** Co-Immunoprecipitation to determine the interaction between Pin1 and Ubc9 in GSCs treated with RO3306 (20 µM) or DMSO for 12 h. **e** Co-immunoprecipitation to determine the interaction between Pin1 and the wild type

Ubc9 or the Ubc9-S71A mutant in 293 T cells. Myc-tagged wild type (WT) Ubc9 or the Ubc9-S71A mutant was overexpressed in 293 T cells along with flag-tagged Pin1. Cell lysates were subjected to immunoprecipitation with anti-flag antibodies followed by immunoblot with anti-myc antibodies. **f** Co-immunoprecipitation to determine the interaction between Pin1 and the wild type Ubc9 or the Ubc9-S71A mutant in 293 T cells after inhibition of CDK1. Cells were treated with RO3306 (20 µM) or DMSO for 12 h. **g** Immunoblot analysis of global protein sumoylation status in GSCs treated with RO3306 (20 µM) or DMSO for 12 h. The blotting experiments were repeated at least three times with biological replicates (**a**–**g**). Source data are provided as a Source Data file.

Consistently, while both sulfopin and RO3306 treatment extended survival of animals bearing intracranial GBMs, mice treated with sulfopin plus RO3306 had the longest survival relative to other groups (Fig. 7d, Supplementary Fig. 5c). Immunofluorescent analyses of the resultant tumors showed that both sulfopin and RO3306 reduced SUMO1-modied sumoylation in tumor cells, while sulfopin plus RO3306 treatment showed the strongest inhibition on SUMO1-sumoylation (Fig. 7e, f, Supplementary Fig. 5d, e). In the meantime, SUMO2/3-modified sumoylation in tumor cells was not affected by

sulfopin or RO3306 (Fig. 7e, f, Supplementary Fig. 5d, e). Moreover, sulfopin or RO3306 treatment alone inhibited tumor cell proliferation marked by Ki67 staining, while combined treatment achieved the severest impact on tumor cell proliferation (Fig. 7g, h, Supplementary Fig. 5f, g). In addition, sulfopin or RO3306 treatment alone induced tumor cell apoptosis as represented by the cleaved caspase-3 staining, while combined treatment further elevated tumor cell apoptosis in vivo (Fig. 7g, h, Supplementary Fig. 5f, g). Finally, sulfopin or RO3306 treatment alone targeted the GSC population in tumor tissues as

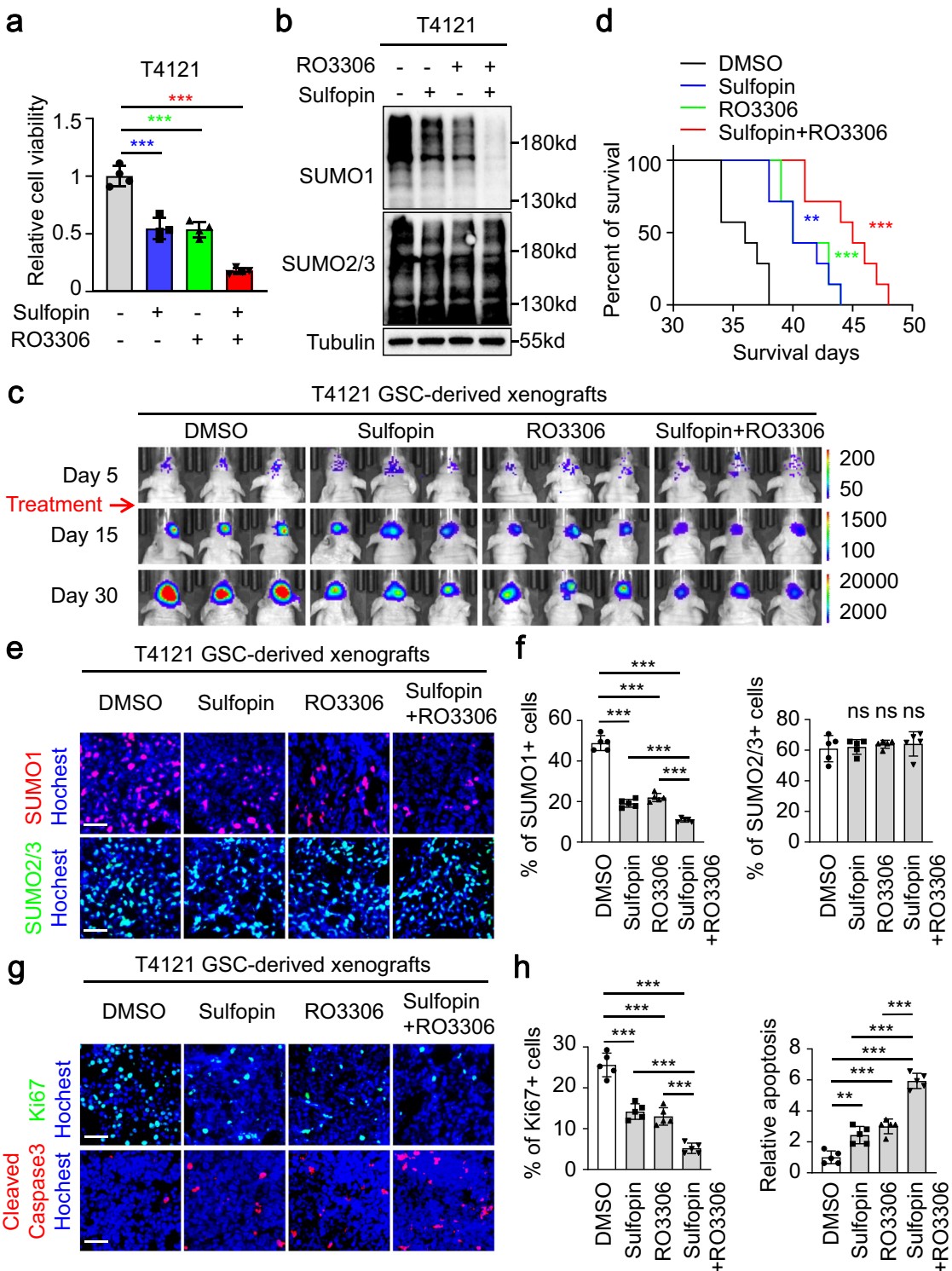

indicated by the reduced OLIG2+ cells, while combined treatment more efficiently impaired GSCs in vivo (Supplementary Fig. 5h, i). Therefore, simultaneous inhibition of Pin1 and CDK1 elevated apoptosis, decreased proliferation, and impacted stemness of tumor cells to delay tumor growth. Of note, immunofluorescent analyses of the GFAP+ astrocytes and NeuN+ neurons in the normal brain regions in GBM-bearing mice showed that sulfopin treatment did not change the intensity or morphology of normal brain cells (Supplementary Fig. 6a–d), indicating that sulfopin had no perceptible toxicity on normal cells. Meanwhile, mass spectrometry analysis of brain tissues from mice treated with sulfopin through intraperitoneal injection proved the capacity of sulfopin to penetrate the blood-brain-barrier and enter the brain tissues (Supplementary Fig. 6e). Collectively, these data show that simultaneous inhibition of Pin1 and CDK1 most effectively disrupts GSCs and mitigates GBM growth.

## Discussion

The current study demonstrates that the peptidyl prolyl cis-trans isomerase Pin1 is deubiquitinated and stabilized by the DUB USP34, which further isomerizes the sole SUMO E2 enzyme Ubc9 to promote SUMO1-modified sumoylation in GSCs. Pin1 is widely overexpressed in malignant cancers[15,40,41], whereas single nucleotide polymorphisms

**Fig. 7 | Simultaneous inhibition of Pin1 and CDK1 suppresses SUMO1-modified sumoylation in GSCs and mitigates GBM growth. a** Cell viability assays of T4121 GSCs treated with the Pin1 inhibitor sulfopin (20 µM), the CDK1 inhibitor RO3306 (20 µM), or sulfopin plus RO3306 for 48 h. GSCs were planted into 96-well plates at 10,000 cells per well. (n = 4 biological replicates; ***, P < 0.001; mean ± s.d.; one-way ANOVA). **b** Immunoblot analysis of global protein sumoylation status in T4121 GSCs treated with the indicated drugs for 24 hours. **c** In vivo bioluminescent imaging analysis of intracranial tumor growth in mice bearing GBM xenografts derived from the T4121 GSCs after treatment with sulfopin (20 mg/kg), RO3306 (20 mg/kg), sulfopin plus RO3306, or DMSO control. One-thousand GSCs transduced with luciferase were implanted into mouse brains through intracranial injection and drug treatment started 5 days post implantation. Intraperitoneal drug administration was performed every other day until the end of the experiment. Representative bioluminescent images on day 5, day 15, and day 30 post-transplantation of GSCs

were shown. **d** Kaplan-Meier survival curves of mice bearing GBMs derived from T4121 GSCs with the indicated treatments. (n = 7 mice for each group; sulfopin vs. DMSO, P = 0.0013; **P < 0.01; ***P < 0.001; two-tailed log-rank test). **e, f** Immunofluorescent analysis (**e**) and statistical quantifications (**f**) of the SUMO1-modified (red) and SUMO2/3-modified (green) protein sumoylation in T4121 GSC-derived xenografts with the indicated treatments. Scale bar, 40 µm (n = 5 tumors for each group; ***P < 0.001; mean ± s.d.; one-way ANOVA). **g, h** Immunofluorescent analysis (**g**) and statistical quantifications (**h**) of Ki67 (green) and cleaved caspase-3 (red) in T4121 GSC-derived xenografts with the indicated treatments. Scale bar, 40 µm (n = 5 tumors for each group; cleaved caspase-3 staining: sulfopin vs. DMSO, P = 0.0013; **P < 0.01; ***P < 0.001; mean ± s.d.; one-way ANOVA). The blotting experiments were repeated at least three times with biological replicates (**b**). Source data are provided as a Source Data file.

associated with decreased Pin1 expression predict a reduced risk for multiple cancers[42]. As a central oncogene, Pin1 emerges as a driver of cancer stem cells in breast, prostate, and brain cancers[4,9,16]. These facts make Pin1 an attractive drug target for cancer treatment, and inducing Pin1 protein degradation may have strong therapeutic effects. Indeed, induced degradation of Pin1 by all-trans-retinoic acid (ATRA) effectively inhibits breast cancers and leukemia[43,44]. By unravelling the mechanisms regulating deubiquitination and stabilization of Pin1 in GSCs, we provide more molecular targets to impair Pin1 protein stability for cancer treatment. On the other hand, hypersumoylation has been shown to promote cancer stem cells in colorectal, breast and brain cancers[4,45,46]. However, the sumoylation machinery has indispensable functions in normal cells and genetic mutations of the key SUMO enzymes rarely occur in mammalian cells[47], indicating that direct targeting of the sumoylation machinery may be toxic. We previously showed that Pin1 is an upstream regulator of global SUMO1-modified sumoylation in GSCs[4]. Here we find that the sole SUMO E2 enzyme Ubc9 is isomerized by Pin1, which selectively promotes the Ubc9 thioester formation with SUMO1 rather than SUMO2/3, resulting in the elevated SUMO1-modified sumoylation. These findings suggest a way to deliberately modulate the sumoylation process for cancer treatment.

We identified USP34 as the DUB responsible for deubiquitination and stabilization of Pin1 proteins. The roles of USP34 in mesenchymal stem cells and DNA damage repair have been reported[48,49], but the involvement of USP34 in tumorigenesis has been rarely studied. Nevertheless, our study clearly demonstrates the tumor supportive role of USP34 in GBMs. We found that the affinity of USP34 to Pin1 is largely enhanced by the Plk1-mediated phosphorylation of Pin1 at S65, which is consistent with a previous report showing that phosphorylation of Pin1 at S65 by Plk1 suppresses Pin1 ubiquitination and increases Pin1 protein stability[22]. Because USP34 is a big protein with a molecular weight of ~400 kDa, we are unable to identify the Pin1 binding site at USP34 at present. In addition to S65, Pin1 proteins can be phosphorylated at multiple serine residues by different kinases including DAPK1, JNK, and MLK3[22,23,50,51]. Although a preferentially high expression of Plk1 was detected in GSCs relative to NSTCs (Fig. 2b), the expression and activation status of the other kinases remained unknown. Inhibition of Plk1 did not abolish serine phosphorylation of Pin1 in GSCs (Fig. 2d), indicating that phosphorylation of serine residues other than S65 may occur, but the exact phosphorylation sites and the functions require further elucidation. Whereas S71 and S138 are associated with the activity of Pin1, the S115 residue phosphorylated by JNK has been reported to stabilize Pin1 protein through inhibiting mono-ubiquitination at K117 in intrahepatic cholangiocarcinoma cells[23]. Interestingly, we identified K117 residue as the poly-ubiquitination site of Pin1 (Fig. 1i–l), but we did not detect the mono-ubiquitination of Pin1 in GSCs, suggesting that ubiquitin-proteasomal degradation of Pin1 proteins may differ in different cancers. Despite the above knowledge about the mechanisms regulating Pin1

ubiquitination, the E3 ubiquitin ligases responsible for Pin1 ubiquitination is not determined yet. However, Pin1 is directly or indirectly complexed with some E3 ubiquitin ligases[8], but the actual E3 ubiquitin ligase of Pin1 needs further investigation.

We found that Pin1 binds to the phosphorylated S71 residue at Ubc9 and isomerizes the pS71-Pro bond (Figs. 5c and 6e). Unlike most known PPIases, Pin1 specifically recognizes the pS/T-Pro motif in substrate proteins. Phosphorylation of the S/T residue not only dramatically slows down the intrinsic isomerization rate of S/T-Pro bonds, but also renders the peptide bonds resistant to the catalytic action of other PPIases[52]. Therefore, isomerization of Ubc9 at pS71-Pro bond should be utterly dependent on the isomerase activity of Pin1. Our study showed that CDK1 phosphorylates Ubc9 at S71, and that isomerization of the pS71-Pro bond by Pin enhances Ubc9 thioester formation with SUMO1, leading to SUMO1-modified hypersumoylation in GSCs. Consistently, previous studies have shown that CDK1-mediated phosphorylation of S71 elevates the activity of Ubc9 and increases the amount of Ubc9-SUMO1 conjugate[38,53,54]. Meanwhile, it has been reported that Akt directly phosphorylates Ubc9 at T35 to promote Ubc9 thioester formation with SUMO1[55]. As T35 is not followed by a proline residue, its phosphorylation should be irrelevant to isomerization. Whereas we have every confidence that Pin1 isomerizes Ubc9 to support GSCs, it is difficult to know if a cis or trans configuration of pS71-Pro bond is favorable for GSC maintenance. Pin1-catalyzed cis-trans isomerization is bi-directional, and either cis or trans configuration could define subsequent molecular events in a content-dependent manner. For example, Pin1-mediated isomerization generates trans configuration at pT-Pro in Cdc25C to promote dephosphorylation[56]. In contrast, Pin1-induced cis configuration at pS-Pro in RNA polymerase II accelerates dephosphorylation[57]. The structural analysis of the Ubc9 thioester with SUMO1 at the presence of Pin1 would be necessary to interpret the conformation of Ubc9 in GSCs.

Our previous studies showed that Pin1 is an upstream regulator of global SUMO1-modified sumoylation in GSCs. Theoretically, it would be a more economic and effective way for Pin1 to act on a pivotal enzyme such as the sole SUMO E1 enzyme SAE1/2 or the sole SUMO E2 enzyme Ubc9 to control the general output of sumoylation. The important clues come from the existence of the S/T-Pro motif and the reports about the regulation of SUMO enzyme activity by phosphorylation. Although potential S/T-Pro motifs are found in the SUMO E1 enzyme subunits SAE1 and SAE2, phosphorylation has not been detected on SAE1, and SAE2 phosphorylation is basically associated with DNA damage response but not sumoylation[58]. These facts make the SAE1/2 less likely to be the Pin1 substrate that controls global sumoylation in GSCs. We found that Pin1-catalyzed isomerization strongly enhances Ubc9 thioester formation with SUMO1 but not SUMO2/3 (Fig. 5g). SUMO1 and SUMO2/3 share less than 50% sequence identity, resulting in different protein structures[59]. Since the pS72-Pro motif of Ubc9 is predicted to be in a pocket responsible for the intermediate covalent interaction between Ubc9 and the SUMO

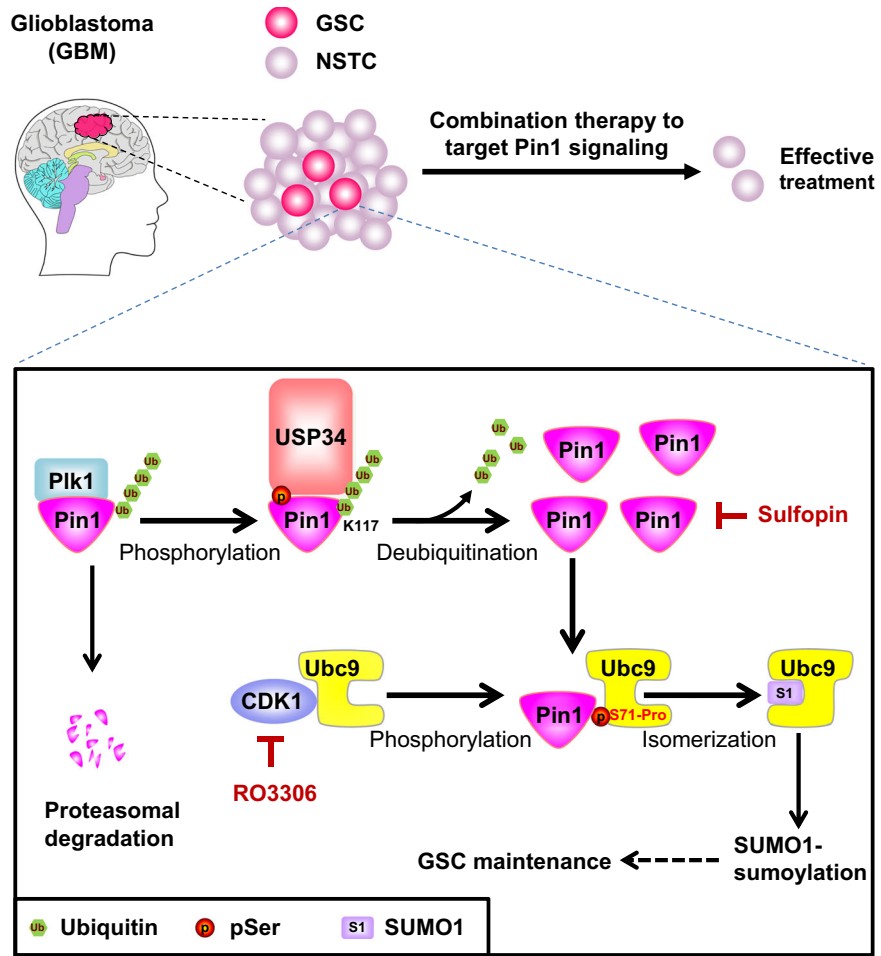

**Fig. 8 | Schematic presentation of targeting Pin1 signaling to disrupt GSCs.** Pin1 is phosphorylated by Plk1 at S65, which facilitates the binding of USP34 to Pin1. The deubiquitinase USP34 removes the poly-ubiquitin chain from the K117 residue and stabilizes Pin1 in GSCs. Pin1 then binds to the phosphorylated S71 site of Ubc9 and catalyzes Ubc9 isomerization to promote Ubc9 thioester formation with SUMO1, which elevates the SUMO1-modified sumoylation. Pin1-catalyzed Ubc9 isomerization relies on the CDK1-mediated phosphorylation of S71 in Ubc9. Simultaneous inhibition of the PPIase activity of Pin1 with sulfopin and the kinase activity of CDK1 with RO3306 most effectively suppresses Pin1 signaling and hypersumoylation to disrupt GSCs.

modifiers[60], Pin1-catalyzed isomerization of Ubc9 may change the configuration of the pocket to enhance the intermediate interaction between Ubc9 and SUMO1 but not the SUMO2/3 protein. However, it is likely that only a small fraction of Ubc9 proteins are subjected to Pin1-catalzyed isomerization that favors the thioester formation with SUMO1, and there are enough Ubc9 proteins left to carry out the SUMO2/3 conjugation. In fact, NSTCs relative to GSCs have much less Ubc9 and Pin1 proteins (Fig. 1a and Supplementary Fig. 4d), but NSTCs have strong SUMO2/3-modified sumoylation equal to GSCs[4]. These facts indicate that SUMO1- and SUMO2/3-modified sumoylation are not functionally counteracting. It is possible that a large portion of Ubc9 proteins, regardless of protein conformation, are not actively participating in catalyzing sumoylation.

It is worth noting that the interaction between Pin1 and Ubc9 may lead to the possible sumoylation of Pin1 by Ubc9. Interestingly, Pin1 is reported to be sumoylated at lysine 6 (K6) and lysine 63 (K63), and the sumoylation is specifically promoted by SUMO1 but not SUMO2/3. Such sumoylation is found to inhibit substrate binding and the phospho-specific PPIase activity of Pin1 in normal cells[61]. However, we found that mutation of K6 and K63 to arginine destabilized the Pin1 protein by inducing its poly-ubiquitination and degradation, which severely impaired the Pin1-mediated upregulation of SUMO1-sumoylation in GSCs (Supplementary Fig. 7a-d), indicating that sumoylation of Pin1 itself may play a unique regulatory role in cancer cells. Besides the potential Ubc9-mediated sumoylation of Pin1,

SUMO1 modification of Plk1 had been reported to increase its stability[62], suggesting a potential regulatory network containing Plk1, Ubc9, CDK1 and Pin1. Nevertheless, we found that inhibition of CDK1 to suppress Ubc9-mediated sumoylation did not reduce the Plk1-mediated phosphorylation of Pin1 (Supplementary Fig. 7e). Meanwhile, although inhibition of Plk1 reduced phosphorylation of Ubc9 at some unknown residues, the interaction between Ubc9 and Pin1 was not interrupted (Supplementary Fig. 7f, g). Taken together, the potential feedback circuits may be of minor importance for Pin1 and Ubc9 mediated global sumoylation in GSCs.

Pin1 has been regarded as an attractive drug target for years. Because the known Pin1 upstream regulators are transcription factors and miRNAs with multiple target genes, it is hard to find a way to specifically suppress Pin1 expression. Alternatively, people look into inhibitors of the PPIase activity of Pin1. A few Pin1 isomerase inhibitors including juglone, PiB, API-1, etc. have shown anti-cancer activities in pre-clinical models of different cancers[4,41]. Nevertheless, the low solubility along with the short half-life largely limit the practical application of these drugs, and people are endeavoring to find more Pin1 inhibitors. Our work suggests that direct impairment of Pin1 protein stability may be an effective way to target Pin1 signaling in cancers. Disruption of USP34 severely reduced Pin1 protein levels (Fig. 1g), suggesting that USP34 inhibitors, although not available so far, would inhibit Pin1 expression. In addition, interfering the binding of Pin1 to USP34 by inhibiting Pin1 phosphorylation with the Plk1

inhibitor SBE 13 downregulated Pin1 expression (Fig. 2c), indicating Plk1 as another target to promote Pin1 degradation. As SBE 13 and most other Plk1 inhibitors have not been reported to penetrate the blood brain barrier, the drug delivery approach may need further optimization[63]. Aside from targeting Pin1, accumulating evidence suggests that suppressing the protein hypersumoylation in cancer stem cells would benefit cancer treatment[4,46,47,55]. This study showed that Pin1 binds to and isomerizes the phosphorylated Ubc9 to promote SUMO1-modified sumoylation, suggesting that suppression of hypersumoylation could be accomplished by simultaneous disruption of Pin1-Ubc9 interaction and the Pin1 PPIase activity, which has been confirmed by the combined treatment with the CDK1 inhibitor RO3306 along with the Pin1 inhibitor sulfopin in orthotopic GBMs (Fig. 7). Considering that both Pin1-catalyzed isomerization and Ubc9-mediated sumoylation have several substrates, simultaneous targeting of Pin1 expression and Ubc9 isomerization may have outstanding anti-cancer effects in multiple cancers. Probably due to the insolubility in water and ethanol, RO3306 has not been applied in clinical trial. Future studies may discover more inhibitors for CDK1 and Pin1 to achieve the combination therapy for cancer treatment.

In summary, our study unravels the molecular mechanisms that stabilize Pin1 protein, promote Ubc9 isomerization, and upregulate SUMO1-modified sumoylation in GSCs (Fig. 8). The discovery of the USP34-catalyzed deubiquitination of Pin1 expands our knowledge about the regulation of the pivotal PPIase Pin1. The finding that Pin1 isomerizes Ubc9 to selectively promote thioester formation with SUMO1 and hence elevates global sumoylation deepens our understanding of the sumoylation process. Last but not least, all the regulatory enzymes discovered in this study, including USP34, Pin1, Plk1, CDK1 and Ubc9, could be effective drug targets for combination therapies to treat GBMs and potentially other malignant tumors.

## Methods

Our research complies with all relevant ethical regulations of the University of Science and Technology of China. All animal protocols were approved by the Animal Research Ethics Committee of the University of Science and Technology of China and animal experiments were performed following the guidelines for the use of laboratory animals. The collection and use of clinical materials were approved by the Medical Research Ethics Committee of The First Affiliated Hospital of the University of Science and Technology of China.

### Human GBM specimens and cells

De-identified GBM surgical specimens were collected from the First Affiliated Hospital of the University of Science and Technology (Anhui, China) in accordance with an Institutional Research Ethics Committee-approved protocol. Informed consent was obtained from all subjects and experiments were approved by the Medical Research Ethics Committee of The First Affiliated Hospital of the University of Science and Technology of China. Participants are not compensated for the involvement in this study. GBM tissues were obtained from patients aged 43–66 years. Sex is not considered in the study design because occurrence of GBM is irrelevant to the gender of patients. The GBM tissues 4404, 1205 and 5425 were from female patients. The GBM tissues 5926, 5840 and 8197 were from male patients. T387, T4121, H2S and T3832 GSCs were kind gifts from Dr. Jeremy Rich (University of Pittsburgh). GSCs were isolated from primary GBMs or xenografts and cultured as previously described[4,64]. In brief, GBM tumors were dissociated and glioma cells were recovered in the stem cell medium (Neurobasal-A medium (Thermo Fisher, A2477501) supplemented with B27 supplement (Thermo Fisher, 12587010), 10 ng/ml EGF, 10 ng/ml bFGF, 2 mM L-glutamine, and 1 mM sodium pyruvate). The isolated cells were then subjected to magnetic-activated cell sorting (MACS) with CD133 microbeads (Miltenyi, 130-097-049) and CD15 microbeads (Miltenyi, 130-046-601) to obtain the GSC population (CD15+/CD133+)

and the NSTC population (CD15−/CD133−). The cancer stem cell phenotypes of the isolated GSCs, including self-renewal, multipotent differentiation, and tumor-initiation were validated by utilizing serial neurosphere formation, induced differentiation, and in vivo limiting dilution assays. The validated GSCs were cultured in the stem cell medium. Alternatively, NSTCs were induced from GSC differentiation by culturing in RPMI 1640 Medium (VivaCell, C3010-0500) with 10% FBS for 7-10 days. For western blots, the DMEM medium for NSTC was changed to the stem cell medium 12 hours before harvest to avoid potential bias from different cell culture systems.

### Intracranial tumor formation and drug treatment

Intracranial transplantation of GSCs to establish orthotopic GBM xenografts was performed as described[4,65]. GSCs were transduced with the indicated shRNAs and firefly luciferase through lentiviral infection. Cells were selected with puromycin (1 mg/mL) for 48 hours after infection. 1,000 to 5,000 cells were then engrafted intracranially in the cerebral cortex at a depth of 2.5–3.5 mm in immunocompromised female mice aged 6–8 weeks (BALB/c nude, SLAC ANIMAL COMPANY) were randomly assigned to experimental groups and animals were maintained until manifestation of neurological signs. Bioluminescent imaging was used to monitor intracranial GBM growth in mice by intraperitoneal injection of D-luciferin (150 mg/kg, Goldbio, LUCK-1G) followed by signal capture with the Spectrum IVIS imaging system (PerkinElmer). For drug treatment, the Pin1 inhibitor sulfopin (SELL-ECK, S9782) and the CDK1 inhibitor RO3306 (Apexbio, A8885) were solved in DMSO and injected into mice intraperitoneally at a dose of 20 mg/kg for both drugs. Drug treatment was performed every other day until the end of the experiments. All animal protocols were approved by the Animal Research Ethics Committee of the University of Science and Technology of China (USTC), and all animal experiments were performed in accordance to the USTC guidelines for the use of laboratory animals. No specific method was used to predetermine sample size. The experiments were not randomized. Only animals with accidental death (for example, due to infection or intracranial injection) were excluded from the data analysis. The investigators were not blinded to allocation during experiments and outcome assessment.

### Plasmid constructs and lentivirus production

shRNAs targeting USP34 (shUSP34#1, TRCN0000038846; shUSP34#2 TRCN0000038848), Pin1 (shPin1#1, also named as shPin1-3′UTR, TRCN0000001033; shPin1#2, TRCN0000001034), Ubc9 (shUbc9-3′UTR, TRCN0000011077), and the shNT control shRNA (SHC002) were purchased from Sigma-Aldrich. The Pin1, Ubc9, and SUMO1 genes were cloned into the pCDH-CMV-MCS-EF1-Puro vector (System Biosciences CD510B-1). For lentivirus production, 293FT cells were transfected with the desired plasmid together with the helper plasmids pCI-VSVG and ps-PAX2. Seventy-two hours after transfection, lentiviral supernatant was collected and passed through a 0.45 mm syringe filter. Cells were then infected with lentiviruses and selected with 1 μg/ml puromycin.

### Cell counting kit-8 (CCK8) assay

Cells were split into 96-well plates at a concentration of 2000 cells per well in 0.2 ml medium. Cell viability was determined on day 1, 3, 5, and 7 by incubating cells with the CCK8 reagent (Vazyme, A311-02) for 2 h followed by detection of OD450. For RO3306 and sulfopin treatment, cells were split at a concentration of 10,000 cells per well and cell viability was determined 48 h post treatment.

### Immunoblot analysis and immunoprecipitation

For immunoblot, cells were lysed in RIPA buffer (50 mM Tris-Cl pH 8.0, 150 mM NaCl, 5 mM EDTA, 1% NP-40, 0.1% SDS, 1 mM Na3VO4, 10 mM NaF, 2 mM DTT, 1 mM PMSF, and phosphatase and protease inhibitor

cocktail (Roche, 11873580001)). Cell lysates were subjected to SDS-PAGE and transferred to PVDF membrane. Membranes were washed with Tris-buffered saline containing 0.1% Tween20 (TBST), blocked with 5% milk for 1 hour, and incubated with primary antibody overnight at 4 °C. The next day, membranes were washed with TBST for 3 times and incubated with horseradish peroxidase conjugated goat anti-rabbit or anti-mouse IgG (1:5000) for 1 hour at room temperature. Membranes were then washed with TBST for 3 times and subjected to enhanced chemiluminescent substrate. Signals were detected with ChemiDoc Imaging System (BIO-RAD). For immunoprecipitation, cells were lysed in the IP Buffer (50 mM Tris-HCl pH 7.5, 150 mM NaCl, 2 mM EDTA, 1.5 mM MgCl2, 1% NP-40, and phosphatase and protease inhibitor cocktail) at 4 °C. After centrifugation at 16,000 g for 10 min, cell supernatants were collected. Approximately 1-3 mg total protein were incubated with the indicated primary antibody along with protein A-conjugated beads (Thermo, 20333), anti-flag antibody-conjugated beads (Abcam, M20018M), or anti-myc antibody-conjugated beads (Abcam, M20030M). The immunoprecipitation system was adjusted to a volume of 1 mL with cold PBS supplemented with 1% NP-40, and the mixture was subjected to constant rotation at 80 rpm overnight at 4 °C. Immunocomplexes were washed three times with ice-cold 0.3% NP-40 in IP buffer, eluted in SDS loading buffer by boiling for 10 min, and analyzed by immunoblot. For co-immunoprecipitation, a fraction of the immunoprecipitation product was used for immunoblot of the prey, and the rest samples were used for detection of the bait protein. Specific antibodies against Pin1 (Invitrogen, PA5-80902, 1:1000; Proteintech, 10495-1-AP, 1:1000), Ubc9 (Abcam, ab33044, 1:1000), USP34 (Santa cruz, sc-100631, 1:500; Bethyl Laboratories, A300-824A, 1:1000), HA (Sigma-Aldrich, 11867423001, 1:2500), tubulin (Sigma-Aldrich, T9026, 1:5000), Flag (Sigma-Aldrich, F1804, 1:2500), CDK1 (Proteintech, 19532-1-AP, 1:1,000), Plk1 (Santa cruz, sc-17783, 1:1000), ubiquitin (Proteintech, 10201-2-AP, 1:1000), pSer (Sigma-Aldrich, 05-1000X, 1:1,000; Santa cruz, sc-81514, 1:500), SUMO1 (Invitrogen, 33-2400, 1:1000; Cell Signaling Technology, 4930S, 1:1000), SUMO2/3 (MBL, M114-3, 1:1000), and SOX2 (Cell Signaling Technology, 3579, 1:2000) were used for immunoblot. Specific antibodies against Pin1 (Invitrogen, PA5-80902) and Ubc9 (Abcam, ab33044) or normal rabbit IgG (Cell Signaling Technology, 2729) were used for immunoprecipitation, and the amount of antibody was 1 µg of antibody per mg of protein.

## In-gel mass spectrometry

Polyacrylamide gel containing the protein bands of interest was cut into 1–1.5 mm$^3$ pieces and placed in a 1.5 ml tube. Samples were collected by spinning the tube and the precipitates were sent to Shanghai Applied Protein Technology Co., Ltd. for analysis. LC-MS/MS analysis was performed on a Q Exactive mass spectrometer (Thermo Scientific) that was coupled to Easy-nLC 1000 liquid chromatograph (Thermo Scientific). MS/MS spectra were searched using MASCOT engine (Matrix Science, London, UK; version 2.2) against a nonredundant International Protein Index arabidopsis sequence database v3.85 (released at September 2011; 39679 sequences) from the European Bioinformatics Institute (http://www.ebi.ac.uk/). For protein identification, the following options were used. Peptide mass tolerance = 20 ppm, MS/MS tolerance = 0.1 Da, Enzyme = Trypsin, Missed cleavage = 2, Fixed modification: Carbamidomethyl (C), Variable modification: Oxidation (M).

## Quantitative PCR

Total RNA was isolated with the EZ-10 Spin Column Total RNA Isolation Kit (BBI, B610583), reverse transcribed with the PrimeScript RT Master Mix (TAKARA, RR036A), and analyzed by quantitative PCR using SYBR Green PCR Master Mix (Biosharp, BL705A) with the LightCycler 96 Real Time PCR instrument (Roche). Samples in triplicates were subjected to two-step real-time RT-PCR analysis. Cycle threshold (Ct) values were

determined automatically by the instrument. The ΔΔCt method was used to calculate relative expression of the target genes by using GAPDH as the internal control. At least three biological repeats were performed for each analysis, whereas a representative result containing three technical replicates were used to generate graphs. Forward primer for USP34: 5′-TCCCGTACCACTTAGACATCTAC-3′. Reverse primer for USP34: 5′-GCTAGTGCGTTATTCCACAGT-3′. Forward primer for Pin1: 5′-TCAGGCCGAGTGTACTACTTC-3′. Reverse primer for Pin1: 5′-TCTTCTGGATGTAGCCGTTGA-3′.

## Immunofluorescent staining

Immunofluorescent staining was performed as described before[4,65]. Surgical human GBM specimens or intracranial xenografts were fixed overnight in 4% PFA at 4 °C, stored in 30% sucrose solution overnight at 4 °C, embedded in OCT at −20 °C, and cryosectioned at a thickness of 7 microns. For staining of SUMO2/3, Ki67, USP34, and Pin1, antigen retrieval was performed by boiling the sections in Tris/EDTA buffer pH 9.0 for 10 min. Sections were then incubated in a PBS solution containing 5% donkey serum plus 0.3% Triton X-100 for 1 h at room temperature for blocking and permeabilization. For SUMO1 staining, a thorough permeabilization was performed by incubating the samples in ice-cold methanol:acetone (1:1) at −20 °C for 5 min. Sections were incubated with primary antibodies (1:200 dilution) in PBS containing 5% donkey serum overnight at 4 °C. After then, the sections were washed with 0.1% TWEEN-20 in PBS and incubated with secondary antibodies (1:1000 dilution) plus Hoechst (1:20,000 dilution) in PBS containing 5% donkey serum for 2 h at room temperature in dark. After final wash with 0.1% Tween-20 in PBS, a coverslip was mount on sections, and the staining was subjected to microscopy. Specific antibodies against Pin1 (Invitrogen, PA5-80902), USP34 (Santa cruz, sc-100631; Bethyl Laboratories, A300-824A), SOX2 (R&D Systems, AF2018), Olig2 (R&D Systems, AF2418), SUMO1 (Cell Signaling Technology, 4940), SUMO2/3 (MBL, M114-3), cleaved caspase-3 (Cell Signaling Technology, 9661 S), Ki67 (Abcam, ab15580), GFAP (BioLegend, 840001), and NeuN (abcam, ab177487) were used for the staining of GBM tumor sections as indicated. All the primary antibodies were used with a dilution of 1:200.

## Cis-trans isomerization assay

The cis-trans isomerization assay was performed as modified from previous studies[12,14,34,35]. Ectopic flag-tagged Pin1 proteins in 293 T were purified by immunoprecipitation with anti-flag antibody (Sigma-Aldrich, F1804, 1 µg of antibody per mg of protein) and protein A-conjugated beads (Thermo, 20333) and elution with 3×flag peptide solution (Beyotime, P9801). Ubc9 peptides (Ubc9-S71: Ac-YPSSPPK-pNA; Ubc9-pS71: Ac-YPS(pS)PPK-pNA) were synthesized by the Yuan-Peptide Biotechnology (Nanjing, China) and dissolved in the reaction buffer (35 mM HEPES pH7.8, 0.4 mg/ml BSA, 2 mM DTT) to a concentration of 1 mM. At the beginning of the cis-trans isomerization assay, Ubc9 peptides (60 µl) were incubated with trypsin (50 µL, 100 µg/mL in 35 mM HEPES pH7.8) (BBI, A003702) on ice for 2 min to completely hydrolyze the prolyl bond at the trans configuration and obtain pure cis isomers. After then, the purified Pin1 proteins (50 µl) or control elutes were added into the reaction system. After re-equilibration, isomerization was processed and aliquots were taken at the indicated time for measurement of the 4-nitroaniline released from the Ubc9 peptides. The absorbance of the released pNA (4-nitroaniline) at 390 nm was determined with BIOTEK Synergy H1. The concentration of 4-nitroaniline was calculated according to standard curves of A390.

## Mice

BALB/c nude mice aged 6–8 weeks were purchased from SLAC ANIMAL COMPANY. Mice were housed at the animal facility of University of Science and Technology of China. Sex was not considered in study

design. Female mice were used in this study to reduce aggression-related injuries.

## Animal studies

All animals were housed at a suitable temperature (22–24 °C) and humidity (40–70%) under a 12/12-h light/dark cycle with unrestricted access to food and water for the duration of the experiment. All animal experiments were conducted with approval from the Animal Research Ethics Committee of the University of Science and Technology of China. For xenograft model, 1000 to 5000 cells were engrafted intracranially into immunocompromised mice (BALB/c nude; SLAC ANIMAL COMPANY) into the cerebral cortex at a depth of 2.5–3.5 mm. The approved protocol permits the tumor growth in mice until the presence of neurological signs such as seizure, impaired movement, tail tone, etc., or for a maximum of 60 days. This study strictly follows the approved protocol. At the end of animal studies, mice were euthanized through carbon dioxide inhalation.

## Statistics and reproducibility

The level of significance was determined by the unpaired two-tailed Student's $t$ test, one-way or two-way analysis of variance (ANOVA) test, or two-tailed log-rank test with $\alpha = 0.05$ (Kaplan–Meier survival curves) with the GraphPad Prism 6 software. Data distribution was assumed to be normal, but this was not formally tested. All quantitative data presented are the mean ± s.d. from at least three samples or experiments per data point. $P < 0.05$ is considered statistically significant. Precise experimental details (number of animals or cells and experimental replication) are provided in the figure legends. Randomization applies to all statistical analyses and the allocation of mice to treatment groups. Data collection and analysis are not performed blind to the operators. Each experiment has at least three biological repeats.

## Reporting summary

Further information on research design is available in the Nature Portfolio Reporting Summary linked to this article.

# Data availability

The mass spectrometry proteomics data generated in this study have been deposited to the ProteomeXchange Consortium via the PRIDE partner repository with the dataset identifier PXD041043 that is publicly available. All data in the article, supplementary information is available. Source data are provided with this paper.

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

## Acknowledgements

We thank Dr. Jeremy N. Rich for kindly providing the cells used in this study. We are grateful to the core facilities at the University of Science and Technology of China (USTC) and the First Affiliated Hospital of USTC for their assistance in flowcytometry and microscopy. This work was supported by the National Natural Science Foundation of China grants Nos. 81972782 and 82173354 (W.Z.), the Anhui Provincial Natural Science Foundation grant No. 2108085J43 (W.Z.), the National Natural Science Foundation of China grant No. 82373087 (A.Z.), the Anhui Provincial Natural Science Foundation grant No. 1908085MH283 (D.L.), and the Key Laboratory of Tumor Immunology and Pathology (Army Medical University), Ministry of Education grant No. 2021jsz709 (D.L.).

## Author contributions

W.Z., X.B., and A.Z. developed the working hypothesis and scientific concept. W.Z., A.Z., and Q.Z. designed the experiments, analyzed the data, and prepared the manuscript. Q.Z. and P.L. performed the experiments and organized the data. D.L. and C.N. provided surgical specimens. H.M., F.L., W.M., C.Y.C., W.W., C.C., and H.W. assisted the experiments. Y.S., X.Y., Y.P., and H.W. helped manuscript preparation and provided scientific input.

## Competing interests

The authors declare no competing interests.
