## [Peer Review File · Nature Communications]

Stabilization of Pin1 by USP34 promotes Ubc9 isomerization and protein sumoylation in glioma stem cellsEditorial Note: Parts of this Peer Review File have been redacted as indicated to maintain the confidentiality of unpublished data.

REVIEWER COMMENTS

Reviewer #1 (Remarks to the Author):

The prolyl isomerase Pin1 regulates protein conformation in many cancers, but we are limited in our understanding of Pin1 protein stability. The authors address the mechanism underlying high Pin1 expression in glioma stem cells. They identify ubiquitin protease USP34 as Pin1 interactor and show that de-ubiquitylation of Pin1 by USP34 enhances its stability by preventing proteasomal degradation. Increased Pin1 stimulates Ubc9 isomerization and subsequent increased SUMO1 conjugation to stimulated tumor growth. Phosphorylation of Pin1 by Plk1 and Ubc9 by CDK1 contribute to this exciting ubiquitylation-phosphorylation-prolyl isomerization-sumoylation post-translational modification axis. Overall, the authors synthesize novel and known facts into an impressive integrated post-translational modification network that contributes to glioma. These results provide novel treatment options for glioblastoma.

Major Comments

1. Given the non-essential role of SUMO1 (Evdokimov E et al. J Cell Sci. 2008;121:4106–4113. Zhang FP et al. Mol Cell Biol. 2008;28:5381–5390), it is puzzling to understand how increased SUMO1 conjugation can lead to enhanced tumour growth. It would be helpful to identify the SUMO1 conjugates by mass spectrometry in this context, ideally in a site-specific manner. Are the known SUMO1 targets increased for SUMO1 conjugation, or is SUMO1 conjugated to other target proteins?
2. The mechanism underlying the preferential conjugation of SUMO1 by Ubc9 upon isomerization by Pin is unclear. It would be helpful to provide mechanistic details for this part of the manuscript.
3. The authors demonstrate that Ubc9 is a substrate for Pin1. However, Pin1 regulation of SUMO1 conjugation could be more complex since the SUMO1-specific protease SENP1 contains 6 S/TP motifs and the SUMO1-specific E3 ligase PIAS4 5 contains S/TP motifs. Stimulating Ubc9, PIAS4 and inhibiting SENP1 could provide a comprehensive Pin1-dependent mechanism to boost SUMO1 conjugation. It would therefore be useful to study whether SENP1 and PIAS4 are also regulated by Pin1.

SENP1 aa sequence:

```
MDDIADRMMDAGEVTLVNHNSVFKTHLLPQTGFPELQSLSDQQILSSRQGHLDERSFTC
STRSAAYNPSYYSDNPSSDSFLGSGDLRTFGQSANGQWRNSTPSSSSSLQKSRNSRSLYL
ETRKTSSGLSNSFAGKSNHHCHVSAYEKSFPIKPVSPSPSWGSCRRSLLSPKKTQRRHVS
TAEETVQEEEREIYRQLLQMVTGKQFTIAKPTTHFPLHLRCLSSSKNTLKDLSLFKNGNSCA
SQIIGSDTSSSGSASILTNQEQLSHSVYLSSTYTPDVAFGSKDSGTLHHPHHHSVPHQPD
NLAASNTQSEGSDSVILLKVKDSQTPTPSSTFFQAELWIKELTSVYDSRARERLRQIEEQKA
LALQLQNQRLQEREHSVHDSVELHLRVPLEKEIPVTVVQETQKKGHKLTDSEDEFPEITEEM
EKEIKNVFRNGNQDEVLSEAFRLTITRKDIQTLNHLNWLNDEIINFYMNMLMERSKEKGLPS
VHAFNTFFFTKLKTAGYQAVKRWTKKVDVFSVDILLVPIHLGVHWCLAVVDFRKKKNITYYDS
MGGINNEACRILLQYLKQESIDKKRKEFDTNGWQLFSKKSQEIPQQMNGSDCGMFAKYA
```

DCITKDRPINFTQQHMPYFRKRMVWEILHRKLL

PIAS4 aa sequence:

MAAELVEAKNMVMSFRVSDLQMLLGFVGRSKSGLKHELVTRALQLVQFDCSPELFFKKIKEL
YETRYAKKNSEAPAPQPHRPLDPLTMHSTYDRAGAVPRTPLAGPNIDYPVLYGKYLNLGRL
PAKTLKPEVRLVKLPFFNMLDELLKPTLVQNNNEKLQESPCIFALTQRQVELIRNSRELQPG
VKAVQVVLRICYSDTSCPQEDQYPPNIAVKVNHSYCSVPGYYPSNKPGEVPEKRPCRPINLT
HLMYLSSATNRITVTWGNYGKSYSVALYLVRQLTSSELLQRLKTIGVKHPELCKALVKEKLR
LDPDSEIATTGVRVSLICPLVKMRLSVPCRAETCAHLQCFDAVFYLMNEKKPTWMCPVCD
KPAPYDQLIIDGLLSKILSECEDEIEYLVDSWCPIRAEEKERSCSPQGAILVLGSPDANGL
LPAPSVNGSGALGSTGGGGPVGSMENGGKPGADVVDLTDSSSSSEDEEEEEEEEEDEDE
EGPRPKRRCPFQKGLVPAC

4. In the discussion section of the manuscript, the authors mention the feedback loop that consists of Pin1 sumoylation to reduce its activity. It would be good to include experiments to strengthen this part. Would the use of a sumoylation-deficient mutant of Pin1 further increase its activity?

5. The resolution of the microscopy pictures is suboptimal and the images appear to be overexposed. Adding high resolution images with correct exposures for 3-5 cells per field would be useful.

6. The authors identify Pin1 as Plk1 substrate and Ubc9 as CDK1 substrate. Since Plk1 is downstream of Ubc9, CDK1 inhibition will also inhibit Plk1. It would be interesting to test whether Pin1 is also directly regulated by CDK1 and whether Ubc9 can be phosphorylated by Plk1.

Minor comments

7. The clinical relevance of RO-3306 is limited due to solubility issues. It is therefore not used in clinical trials. It is good to mention this in the discussion section of the manuscript.

8. Please note that the correct term is 'Hoechst', not 'Hochest'.

9. Typo Fig 6G: SUOM2

Reviewer #2 (Remarks to the Author):

In this manuscript, Zhu et al., aim to decipher the mechanism responsible for the high expression of the prolyl isomerase Pin1 that is associated with SUMO1 hypersumoylation in glioma stem cells (GSC). The authors, using classical biochemistry methodologies, follow a step by step approach to clearly demonstrate for the first time that USP34 interacts with Pin1 leading to its stabilization, induction of Ubc9 isomerization and SUMO1 hypersumoylation in GSCs. They began their study by showing for the that Pin1 interacts with USP34 leading to Pin1 deubiquitination and stabilization. Next, they studied how this interaction is regulated and the role of USP34 in GSC maintenance and in vivo glioma growth. The authors followed up by bridging USP34 role as Pin1 deubiquitinase with SUMO1 hypersumoylation in GSC. Next, the authors focused on Pin1 and its function as isomerase, demonstrating how Pin1 by promoting Ubc9 isomerization, facilitates Ubc9-SUMO1 thioester bond and SUMO1

hypersumoylation. Since CDK1 has been already shown to phosphorylates Ubc9, the authors tested the hypothesis that CDK1 might regulate the interaction of Ubc9 and pin1. Finally, the authors evaluated the co-inhibition of CDK1 and Pin1 with commercially available inhibitors in vitro on GSCs viability and maintenance and in vivo in a GBM model. The findings of the paper are bringing new light on the mechanism leading to SUMO1 hypersumoylation in GBM which is extremely relevant since it has been well documented that SUMO1 play a major role in GBM formation and growth. In addition this study may offer new therapeutic strategy in GBM treatment.

Comments:

Overall, this is a well-written, well-prepared manuscript with a lot of solid data provided. I have only one concern that maybe due to a mistake in the method writing. It seems based on the method section that all the IP for ubiquitination analysis were done in mild IP buffer (basically 1% NP-40). When analyzing the ubiquitination of any protein it is necessary to lysed in denaturing buffer to remove any interacting protein from the one studied. The lysate is then diluted to allow the IP. If as described in this paper and method, the IP was done in mild buffer, the authors cannot guarantee that the ubiquitination observed (for example in Fig1B, 1H) is really Pin1 ubiquitination and not one of the proteins it interacts with. Pin1 by nature as a peptidyl prolyl cis-trans isomerase interacts with numerous proteins including some for which Pin1 isomerization induced ubiquitination. Not IP'ing in denaturing buffer weakened the interpretation of the result.

Minor:

In the introduction line 142, the authors cited PIAS 1-4 as the E3 ligase. They should rephrase the sentence since PIAS1-4 are not the only SUMO E3-ligases, others such as Ran-BP2, TRIM.

Reviewer #3 (Remarks to the Author):

In this manuscript, the authors studied the role of Pin1 in the hypersumoylation in glioma stem cells (GSC) and how this is essential for GSC stemness in vitro and tumor progression for in vivo PDX models. The results derived from this work are exciting since the authors depicted with a high level of detail how Pin1 gets stabilized in GSCs and how Pin1-mediated isomerization of Ubc9 promotes SUMO1 conjugation in GSC, promoting tumorigenesis. This work sheds some light on the understanding of the mechanisms of the high sumoylation in GBM. In terms of therapeutic approach, although targeting Pin1 and CSK1 are not novel strategies, the combination of both targeting the same pathways is interesting.

In general, the results supported the conclusions and claims made by the authors, although I considered some revisions necessary. Also, the methodology used here adjusts to the questions that the authors aim to answer, and, in general, they are very well explained in the section Material and Methods.

The points that I consider need to be revised are the following:

1. NSTC has not been defined. Please review all the acronyms and ensure that they have been defined for the first time they appear in the text.
2. In general, the immunofluorescence images are very saturated, and it is difficult to

distinguish any detail of the cells. Also, nuclei look larger than the cells in some images. In some cases, it is difficult to assume the claims of the authors due to the image quality. Also, it might be because of the conversion, but it looks like the contrast of those images has dramatically increased. Please, include better images.

Also, In figure 4D, I would include NSTCs to see if SUMO1 correlates with USP34 in non-stem cells as well.

3. In the point "Simultaneous inhibition of Pin1 and CDK1 suppresses SUMO1-modified sumoylation in GSCs and mitigates GBM growth", the authors suggest that apoptosis due to inhibition of Pin1 and CDK1 could be behind the slowdown in tumor growth. However, I am unsure whether the apoptosis shown in that image is enough. Additionally, how did the authors identify that the region imaged is actually the tumor?

4. The authors mention in the discussion that inhibition of global sumoylation could be potentially toxic for the cells. Therefore, targeting Pin1, which is the upstream regulator of SUMO1-conjugation could be a better therapeutic strategy for GBM. I wonder whether the treatment with Sulfopin induces toxicity in normal cells, such as neurons or primary astrocytes and whether they observed toxicity on mice during in vivo experiments.

5. Although the authors intend to correct the bias associated with the different cell culture media (FBS for NSTC), I am wondering whether 12h in the same medium is enough. Do the authors have tried to assess the ubiquitination of Pin1 having GSCs in FBS or in NBM to see whether the FBS is affecting the protein stability and expression?

6. The authors mention that most Pin1 inhibitors are unable to pass through BBB. Could they please provide any reference or experimental evidence that Sulfopin is actually able to reach the brain since the treatment is through intraperitoneal injections?

Response to Reviewers' comments

Manuscript NCOMMS-23-12195-T

Reviewer #1

Reviewer #1: *The prolyl isomerase Pin1 regulates protein conformation in many cancers, but we are limited in our understanding of Pin1 protein stability. The authors address the mechanism underlying high Pin1 expression in glioma stem cells. They identify ubiquitin protease USP34 as Pin1 interactor and show that de-ubiquitylation of Pin1 by USP34 enhances its stability by preventing proteasomal degradation. Increased Pin1 stimulates Ubc9 isomerization and subsequent increased SUMO1 conjugation to stimulated tumor growth. Phosphorylation of Pin1 by Plk1 and Ubc9 by CDK1 contribute to this exciting ubiquitylation-phosphorylation-prolyl isomerization-sumoylation post-translational modification axis. Overall, the authors synthesize novel and known facts into an impressive integrated post-translational modification network that contributes to glioma. These results provide novel treatment options for glioblastoma.*

Response: We thank the reviewer for the positive comments on our manuscript. We have performed a large number of additional experiments to address the reviewer's concerns and further validate our main conclusions. Some of the new data with a close relevance to the main conclusions have been included in the revised manuscript. We believe that the revised manuscript has addressed the concerns from the reviewer and strengthened the main conclusions.

Reviewer #1 (Major Comments): *1. Given the non-essential role of SUMO1 (Evdokimov E et al. J Cell Sci. 2008;121:4106–4113. Zhang FP et al. Mol Cell Biol. 2008;28:5381–5390), it is puzzling to understand how increased SUMO1 conjugation can lead to enhanced tumour growth. It would be helpful to identify the SUMO1 conjugates by mass spectrometry in this context, ideally in a site-specific manner. Are the known SUMO1 targets increased for SUMO1 conjugation, or is SUMO1 conjugated to other target proteins?*

Response: We thank the reviewer for raising this important question. We agree with the reviewer that a comprehensive analysis of SUMO1-sumoylated proteins in GSCs is required for the in-depth understanding of the critical role of the increased SUMO1 conjugation in GSCs. As suggested by the reviewer, we performed mass spectrometry to analyze the increased SUMO1 conjugations in GSCs relative to NSTCs. Cell lysates from T387 GSCs and the paired NSTCs were immunoprecipitated with anti-SUMO1 antibodies, and the product proteins were subject to LC-MS/MS (Fig. R1A). The results revealed 336 proteins that had a more than 3-fold enrichment in GSCs relative to NSTCs. KEGG analysis of these 336 proteins enriched in GSCs identified 10 pathways with various important functions that may regulate tumor

growth (Fig. R1B). Interestingly, we found that SUMO1 conjugation may preferentially occur on several carbon metabolism-associated genes in GSCs (Fig. R1C), suggesting a potential important role of SUMO1 conjugation in metabolic reprogramming in GSCs and tumor growth. Of note, our mass spectrometry may detect both sumoylated proteins and their interacting proteins, and further validation is required to determine the actual SUMO1-sumoylation on the substrate proteins.

When it comes to the identification of sumoylation sites on proteins, the methods need several optimizations. Expression of ectopic epitope-tagged SUMO proteins are required to facilitate the enrichment of sumoylated proteins. In addition, an artificial tryptic site needs to be introduced into the ectopic SUMO protein through substitution of a specific residue with an arginine, which not only shortens the SUMO chains but also generates a signature peptide with a di-glycine remnant attached to the lysine residue on substrate proteins after tryptic digestion. The sumoylation sites then could be identified by analysis of the abovementioned signature peptides detected in mass spectrometry. Considering the complex methodology, we do not aim to determine the sumoylation sites at the current stage.

Noticeably, mass spectrometry study of protein sumoylation in GSCs is challenged by the low abundance of sumoylated proteins. Our previous studies showed that the well-known sumoylated protein PML has a stronger SUMO1-modification in GSCs relative to NSTCs (*Neuro Oncol.* 2020 Dec 18;22(12):1809-1821. doi: [10.1093/neuonc/noaa150](https://doi.org/10.1093/neuonc/noaa150)). We found that the SUMO1-modification of PML enhances its interaction with the stem cell transcriptional factor c-Myc and in turn stabilizes c-Myc to promote GSC maintenance. However, although PML proteins have much stronger SUMO1 conjugation in GSCs relative to NSTCs, total PML protein levels are much lower in GSCs than in NSTCs (Fig. R1D, originally Fig. 2C from *Neuro Oncol.* 2020 Dec 18;22(12):1809-1821.). Probably because of the low abundance of PML in GSCs, our mass spectrometry analysis did not detect PML with the current settings.

In summary, through mass spectrometry analysis, we detected several new SUMO1-sumoylated proteins or their partners that are preferentially expressed in GSCs relative to NSTCs. Discovery of these proteins indicated a role of sumoylation in metabolic reprogramming that may be critical for tumor growth. SUMO1 conjugation may also increase on known targets such as PML to support GSCs, but the studies on the known sumoylated targets may be impeded by the low abundance of these proteins. Optimizations of mass spectrometry are required to determine the sumoylation sites and to discover the sumoylated proteins of low abundance. As these preliminary mass spectrometry data need further validation and the related ongoing projects in our lab are beyond the scope of this manuscript, we would like to include these data in the rebuttal letter but not the revised manuscript.

Rebuttal Figure R1

[Redacted]

Figure R1. (A) Lysate of T387 GSCs and paired NSTCs were subjected to immunoprecipitation with anti-SUMO1 antibodies. The SUMO1-modified sumoylation in GSCs and NSTCs was determined by immunoblot of input proteins (left). The immunoprecipitated proteins were separated by gel electrophoresis and stained with coomassie blue (right) before LC-MS/MS analysis.

(B) Candidate proteins with SUMO1-sumoylation preferentially in GSCs were determined with a threshold of 3 times more peptide abundance detected in GSCs relative to NSTCs in mass spectrometry. A total of 336 candidate proteins were obtained and their functions were analyzed according to the KEGG enrichment by using the R package ClusterProfiler. The top 10 pathways with statistical significance were visualized with dotplot diagram. The horizontal generatio parameter of the diagram stands for the ratio of the enriched gene numbers to the total input gene number. The sizes of the dots represent the number of genes, while the color intensity reflects the p-value.

(C) Table of the carbon metabolism-related proteins identified by the mass spectrometry that may have SUMO1 conjugation preferentially in GSCs.

(D) Immunoblot of SUMO1 sumoylation of PML proteins in GSCs and NSTCs. Cell lysate was subjected to immunoprecipitation with anti-PML antibody.

Reviewer #1 (Major Comments):2. The mechanism underlying the preferential conjugation of SUMO1 by Ubc9 upon isomerization by Pin1 is unclear. It would be helpful to provide mechanistic details for this part of the manuscript.

Response: We thank the reviewer for pointing out this important issue. As a matter of fact, we don't think that there is a preferential conjugation of SUMO1 by Ubc9 upon isomerization by Pin1. If isomerization of Ubc9 led to a preferential conjugation of SUMO1, then preventing Ubc9 isomerization would decrease SUMO1 conjugation but elevate SUMO2/3 conjugation. However, disruption of Pin1 and the consequent Ubc9 isomerization dramatically reduced the formation of the Ubc9-SUMO1 conjugate, whereas negligible change was detected on the Ubc9-SUMO2/3 conjugate (Fig. R2A, equals to Fig. 5G and Supplementary Fig. 4G in the revised manuscript), suggesting that Pin1-mediated isomerization promoted SUMO1 conjugation on Ubc9 but did not disturb SUMO2/3 conjugation in general. In line with this observation, the Ubc9 S71A mutant, whose conjugation with SUMO1 was dramatically impeded (Fig. R2B, equals to Fig. 5I in the revised manuscript), largely lost the capacity to promote global SUMO1 sumoylation but did not further enhance SUMO2/3 sumoylation when compared with the wildtype Ubc9 (Fig. R2C, equals to Fig. 5J in the revised manuscript).

The seemingly parallel conjugations of SUMO1 and SUMO2/3 to Ubc9 may be ascribed to the adequate supply of Ubc9 in tumor cells. Levels of Ubc9 proteins were low in NSTCs relative to GSCs (Fig. R2D, equals to Supplementary Fig. 4D in the revised manuscript), but high SUMO2/3 sumoylation was detected at a similar level in GSCs and NSTCs (Fig. R2E, originally Fig. 1B from *Neuro Oncol.* 2020 Dec 18;22(12):1809-1821.), suggesting that a little amount of Ubc9 would be sufficient to execute SUMO2/3 conjugations. Given that SUMO1 sumoylation in general was weaker than SUMO2/3 sumoylation, it is likely that not much Ubc9 protein is required for SUMO1 conjugation. In addition, immunoblot detection of the un-conjugated Ubc9 protein needed a very short exposure, while detection of both Ubc9-SUMO1 and Ubc9-SUMO2/3 conjugates required a long exposure, suggesting that not much Ubc9 protein was conjugated with SUMO modifier proteins. Therefore, Pin1-mediated isomerization may occur on a very small proportion of the Ubc9 protein pool, which would enhance the SUMO1 conjugation without disturbing the SUMO2/3 conjugation.

On the other hand, our data clearly demonstrated that Pin1-mediated isomerization of Ubc9 promoted the formation of the Ubc9-SUMO1 conjugates and hence upregulated global SUMO1-modified sumoylation. Although we currently don't have data concerning the mechanism underlying the enhanced SUMO1 conjugation to the isomerized Ubc9, we do find a clue from a previous report which predicted a pocket on the surface of the human Ubc9 crystal structure (*J Chem Inf Model.* 2014 Oct 27;54(10):2784-93. doi: 10.1021/ci5004015.). The pocket composed of the important catalytic residue C93 and several flanking residues could be the major contact region for intermediate covalent interaction between SUMO and Ubc9 (Fig. R2F, originally Fig. S1B from *J Chem Inf Model.* 2014 Oct 27;54(10):2784-93.). Interestingly, the

isomerization motif on Ubc9, namely the pS71-P72, is localized in this predicted pocket. Therefore, the isomerization of the pS71-P72 motif may probably affect the structure of the pocket that is critical for the interaction between SUMO and Ubc9, thus enhancing the conjugation of SUMO1 to Ubc9.

We have discussed the effects of Ubc9 isomerization on SUMO1 and SUMO2/3 conjugation (Line 545-552, Page 19) in the Discussion section. We now included the description of the potential structural change of the pocket responsible for Ubc9-SUMO interaction upon Ubc9 isomerization in the Discussion section (Line 541-545, Page 19) in the revised manuscript.

Rebuttal Fig. R2

[Redacted]

Figure R2. (A) Immunoblot analysis of the Ubc9-SUMO1 thioester (Ubc9~SUMO1) and Ubc9-SUMO2/3 thioester (Ubc9~SUMO2/3) formation in T387 (left) and T4121 (right) GSCs after disruption of Pin1. GSCs were infected with shPin1 lentiviruses and cell lysates were subjected to immunoprecipitation with anti-Ubc9 antibodies. The Ubc9-SUMO1 and Ubc9-SUMO2/3 conjugates were detected by immunoblot with anti-SUMO1 and anti-SUMO2/3 antibodies. DTT and β -ME were added into protein input but not the immunoprecipitation products. Silencing Pin1 reduced the amount of the Ubc9-SUMO1 conjugate, indicating the disruption of the Ubc9-SUMO1 thioester. Meanwhile, disruption of Pin1 showed negligible effects on the Ubc9-SUMO2/3 conjugate.

(B) Immunoblot analysis of the Ubc9-SUMO1 thioester (Ubc9~SUMO1) formation in 293T cells expressing ectopic SUMO1 and wild type or mutant Ubc9. Myc-tagged wild type Ubc9 or the Ubc9-S71A mutant was transfected into 293T cells along with HA-tagged SUMO1. Cell lysates were subjected to immunoprecipitation with anti-myc antibodies. The Ubc9-SUMO1 conjugate was detected by immunoblot with anti-SUMO1 antibodies. DTT and β -ME were added into protein input but not the immunoprecipitation products. The band of the Ubc9-SUMO1 conjugate was much stronger in cells expressing the wild type Ubc9 relative to those expressing the Ubc9-S71A mutant, indicating the importance of the S71 site at Ubc9 for the Ubc9-SUMO1 thioester formation.

(C) Immunoblot analysis of global protein sumoylation status in T387 GSCs expressing ectopic wild type Ubc9 or the Ubc9-S71A mutant. GSCs were transduced with the wild type Ubc9 or the Ubc9-S71A mutant through lentiviral infection. The cells were further infected with lentiviruses carrying an shRNA targeting the 3'-untranslated region (3'-UTR) of Ubc9 (shUbc9-3'UTR) to silence the expression of endogenous Ubc9. Cell lysates were subjected to immunoblot analysis with indicated antibodies. Much weaker SUMO1-modified protein sumoylation was detected in cells expressing the Ubc9-S71A mutant relative to those expressing the wild type Ubc9, indicating the importance of Pin1-catalyzed isomerization of Ubc9 for global SUMO1-modified sumoylation in GSCs. Meanwhile, SUMO2/3-modified sumoylation levels were similar in cells expressing the wild type Ubc9 or the Ubc9-S71A mutant.

(D) Immunoblot analysis of Ubc9 protein levels in GSCs and NSTCs. GSCs had stronger Ubc9 expression relative to NSTCs.

(E) Immunoblot of global SUMO2/3 sumoylation levels in GSCs and NSTCs.

(F) Computational fragment mapping by FTMap on Ubc9 crystal structure. Only top ranked FTMap consensus site is shown for clarity.

Reviewer #1 (Major Comments):3. *The authors demonstrate that Ubc9 is a substrate for Pin1. However, Pin1 regulation of SUMO1 conjugation could be more complex since the SUMO1-specific protease SENP1 contains 6 S/TP motifs and the SUMO1-specific E3 ligase PIAS4 5 contains S/TP motifs. Stimulating Ubc9, PIAS4 and inhibiting SENP1 could provide a comprehensive Pin1-dependent mechanism to boost SUMO1 conjugation. It would therefore be useful to study whether SENP1 and PIAS4 are also regulated by Pin1.*

Response: We are grateful for these important suggestions raised by the reviewer. The presence of the S/TP motifs in SENP1 and PIAS4 indicates potential interactions between these enzymes and Pin1. To clarify this issue, we performed immunoprecipitation assays to determine the possible interaction between Pin1 and SENP1 or PIAS4. Immunoprecipitation with anti-Pin1 antibodies did not pull-down SENP1 or PIAS4 in T387 or T4121 GSCs (Fig. R3A). Furthermore, immunoprecipitation with anti-SENP1 antibodies followed by immunoblot with anti-Pin1 antibodies detected no interaction between SENP1 and Pin1 (Fig. R3B). We did not do immunoprecipitation with anti-PIAS4 antibodies because we did not have

an antibody well enough for immunoprecipitation. However, these data strongly indicate that Pin1 may not interact with SENP1 or PIAS4 in GSCs.

Pin1 specifically binds to and isomerizes the phosphorylated S/TP motifs on substrate proteins. We thus checked the literatures for the phosphorylation on SENP1 and PIAS4. So far, no literature has described the phosphorylation of the S/TP motifs on SENP1 (T102, S157, S170, T276, T328, and T330) or PIAS4 (S51, T99, S162, T169, and S416), which further excludes the possible direct regulation of SENP1 and PIAS4 by Pin1. Finally, we checked the levels of SENP1 and PIAS4 in GSCs and paired NSTCs to explore their potential regulatory roles in SUMO1-sumoylation in GSCs. Immunoblot analyses showed that SENP1 and PIAS4 have similar protein levels in GSCs and NSTCs (Fig. R3C), suggesting that these enzymes are unlikely involved in the upregulation of SUMO1-sumoylation in GSCs. These data further underscore the importance of the Pin1-mediated isomerization of Ubc9 on SUMO1-sumoylation in GSCs.

Rebuttal Fig. R3

[Redacted]

Figure R3. (A) Co-immunoprecipitation to determine the interaction between endogenous Pin1 and SENP1 or PIAS4 in GSCs. Cell lysate was subjected to immunoprecipitation with anti-Pin1 antibodies or control IgG followed by immunoblot with antibodies against SENP1 or PIAS4. No interaction between Pin1 and SENP1 or PIAS4 was detected in GSCs.

(B) Co-immunoprecipitation to determine the interaction between endogenous Pin1 and SENP1 in GSCs. Cell lysate was subjected to immunoprecipitation with anti-SENP1 antibodies or control IgG followed by immunoblot with anti-Pin1 antibodies. No interaction between Pin1 and SENP1 was detected in GSCs.

(C) Immunoblot analysis of SENP1 and PIAS4 protein levels in GSCs and NSTCs. SENP1 and PIAS4 expression had no difference in GSCs and NSTCs.

Reviewer #1 (Major Comments):4. In the discussion section of the manuscript, the authors mention the feedback loop that consists of Pin1 sumoylation to reduce its activity. It would be good to include experiments to strengthen this part. Would the use of a sumoylation-deficient mutant of Pin1 further increase its activity?

Response: We thank the reviewer for this constructive suggestion. It has been reported that removal of the SUMO1-modifiers from the K6 and K63 residues of Pin1 by SENP1 leads to the stabilization of the Pin1 protein and the upregulation of Pin1 activity (*Cancer Res.* 2013 Jul 1;73(13):3951-62. doi: 10.1158/0008-5472.CAN-12-4360.). Based on this report, we proposed that Pin1 promotes Ubc9-mediated global sumoylation, which may lead to SUMO1 modification of Pin1 itself to reduce the stability and activity of Pin1, resulting in a negative feedback loop. To test this hypothesis, we constructed the sumoylation-deficient Pin1-2KR mutant by replacing the K6 and K63 lysine with arginine. We assumed that Pin1-2KR should have a higher expression relative to the wildtype Pin1 when introduced into GSCs. To the opposite, immunoblot detected a much lower expression of Pin1-2KR than wildtype Pin1 in T4121 and T387 GSCs (Fig. R4A, equals to Supplementary Fig. 7A in the revised manuscript), indicating that abolishing the sumoylation on K6 and K63 may impair the stability of Pin1 in GSCs. Indeed, treatment with the proteasome inhibitor MG132 markedly restored protein expression of the Pin1-2KR mutant in GSCs (Fig. R4B, equals to Supplementary Fig. 7B in the revised manuscript). Furthermore, ubiquitination assays revealed a strong poly-ubiquitination of Pin1-2KR, which was not observed in the wildtype Pin1 in GSCs (Fig. R4C, equals to Supplementary Fig. 7C in the revised manuscript). On the other hand, whereas ectopic expression of the wildtype Pin1 markedly elevated global SUMO1 sumoylation, overexpression Pin1-2KR only slightly increased SUMO1-sumoylation in GSCs (Fig. R4D, equals to Supplementary Fig. 7C in the revised manuscript), which could be ascribed to either the low expression levels or the potential reduced activity of the Pin1-2KR mutant. Taken together, these data strongly suggest that sumoylation of Pin1 at K6 and K63 may stabilize Pin1 proteins in GSCs, whereas the impact of sumoylation on the activity of Pin1 remains an open question. Considering that the previous study about the destabilization of Pin1 upon sumoylation was carried out in normal cells such as NIH3T3 and MCF10A, our data suggest that the role of sumoylation in the regulation of Pin1 may differ in cancer and normal cells. We have added these data in the revised manuscript and modified the Discussion section (Line 556-561, Page 19-20) accordingly.

Rebuttal Fig. R4

[Redacted]

Figure R4. (A) Immunoblot analysis of the protein levels of the wildtype Pin1 (Pin1-WT) and the sumoylation deficient Pin1 mutant (Pin1-2KR) in which the K6 and K63 residues were substituted with arginine. Pin1-WT or Pin1-2KR was introduced into GSCs through lentiviral infection and cells were lysed 72 hours post infection. Much weaker Pin1-2KR relative to Pin1-WT expression was detected in GSCs.

(B) Immunoblot analysis of Pin1-2KR protein levels in GSCs treated with MG132 (10 μ M) or DMSO for 6 hours. MG132 treatment restored Pin1-2KR protein expression in GSCs.

(C) Immunoblot analysis of the poly-ubiquitination of the ectopic Pin1-WT and Pin1-2KR proteins in GSCs. Pin1-WT or Pin1-2KR was introduced into GSCs through lentiviral infection and cells were lysed 72 hours post infection. GSCs were treated with MG132 (10 μ M) for 6 hours before harvest for ubiquitination assays. Cell lysate was subjected to immunoprecipitation with anti-Flag antibodies followed by immunoblot with anti-Ub antibodies. Strong poly-ubiquitination of Pin1-2KR proteins was detected, but negligible poly-ubiquitination of Pin1-WT proteins appeared in the GSCs.

(D) Immunoblot analysis of global protein sumoylation status in T4121 and T387 GSCs expressing ectopic Pin1-WT or Pin1-2KR proteins. GSCs were lysed 72 hours post lentiviral infection. Much weaker SUMO1-modified protein sumoylation was detected in cells expressing Pin1-2KR relative to those expressing Pin1-WT. Meanwhile, SUMO2/3-modified sumoylation levels were similar in cells expressing Pin1-WT or the Pin1-2KR mutant.

Reviewer #1 (Major Comments):5. *The resolution of the microscopy pictures is suboptimal and the images appear to be overexposed. Adding high resolution images with correct exposures for 3-5 cells per field would be useful.*

Response: We apologize for the unsatisfied resolution of the microscopy pictures. We have provided new images with better resolution in the revised manuscript. In addition, we cropped and enlarged a section of 3-5 cells for each new image. These

pictures clearly demonstrate the co-expression of USP34 and Pin1 in GBMs (Fig. R5A and R5B, equal to Fig. 1E and Supplementary Fig. 1C in the revised manuscript), and the co-staining of USP34 with SUMO1 but not SUMO2/3 (Fig. R5C and R5D, equal to Fig. 4D and Supplementary Fig. 3D and 3E in the revised manuscript). We believe that these new images have strengthened our main conclusions.

Rebuttal Fig. R5

[Redacted]

[Redacted]

Figure R5. (A and B) Representative images of immunofluorescent analysis of USP34 (red) and Pin1 (green) in human primary GBMs. Frozen sections of human GBMs were immunostained with antibodies against USP34 and Pin1, and counterstained with Hoechst to show nuclei (blue). The majority of USP34+ cells were positively stained for Pin1 in human GBMs. Scale bar, 20 μ m.

(C and D) Representative images of immunofluorescent analyses of USP34 (red) and SUMO1 or SUMO2/3 (green) in human primary GBMs. Frozen sections of human GBMs were immunostained with antibodies against USP34 and SUMO1 or SUMO2/3, and counterstained with Hoechst to show nuclei (blue). The majority of SUMO1+ cells were positively stained for USP34 in human GBMs, whereas only about half of SUMO2/3+ cells had USP34 staining. Scale bar, 10 μ m.

Reviewer #1 (Major Comments):6. *The authors identify Pin1 as Plk1 substrate and Ubc9 as CDK1 substrate. Since Plk1 is downstream of Ubc9, CDK1 inhibition will also inhibit Plk1. It would be interesting to test whether Pin1 is also directly regulated by CDK1 and whether Ubc9 can be phosphorylated by Plk1.*

Response: We thank the reviewer for raising this important question. SUMO1 modification of Plk1 had been reported to increase its stability (Cell Rep. 2017 Nov 21;21(8):2147-2159. doi: 10.1016/j.celrep.2017.10.085.). Therefore, Plk1 may be at the downstream of Ubc9 and inhibition of Ubc9-mediated sumoylation by the CDK1 inhibitor may impair Plk1 stability and activity, leading to the decrease of Pin1 phosphorylation and its destabilization. To test this possibility, GSCs were treated with the CDK1 inhibitor RO3306 and phosphorylation status of Pin1 was determined through immunoblot. We found that RO3306 treatment showed no perceptible impact on the phosphorylation of Pin1 (Fig. R6A, equals to Supplementary Fig. 7E in the revised manuscript), excluding the possible regulation of Pin1 phosphorylation by CDK1.

We also tested whether phosphorylation of Ubc9 may be regulated by Plk1. Surprisingly, treatment with the Plk1 inhibitor SBE 13 markedly reduced

phosphorylation of Ubc9 in GSCs (Fig. R6B, equals to Supplementary Fig. 7F in the revised manuscript). Ubc9 contains several serine residues that may be subject to Plk1-catalyzed phosphorylation. Because this manuscript is focused on the binding of Pin1 to Ubc9 and the sequent isomerization of Ubc9, we investigated whether the Plk1-catalyzed phosphorylation of Ubc9 may affect the interaction between Ubc9 and Pin1. Co-immunoprecipitation analysis showed that treatment with the Plk1 inhibitor SBE13 did not affect the interaction between Ubc9 and Pin1 in GSCs (Fig. R6C, equals to Supplementary Fig. 7G in the revised manuscript). Therefore, Plk1 has the capacity to phosphorylate Ubc9, but such phosphorylation seems to be irrelevant to the Pin1-catalyzed isomerization of Ubc9. The exact phosphorylation site(s) on Ubc9 protein and the downstream signaling in response to Plk1 activation remain to be investigated, but these are beyond the scope of this manuscript.

We have included the above data along with the corresponding discussions (Line 561-569, Page 20) in the Discussion section in the revised manuscript.

Rebuttal Fig. R6

[Redacted]

Figure R6. (A) Immunoblot analysis of the serine phosphorylation (pSer) status of Pin1 in GSCs after inhibition of CDK1. Cells were treated with RO3306 (20 μ M) or DMSO for 12 hours. Cell lysates were subjected to immunoprecipitation with IgG or anti-Pin1 antibodies followed by immunoblot with anti-pan-pSer antibodies. Inhibition of CDK1 did not change pSer of Pin1 in GSCs.

(B) Immunoblot analysis of the serine phosphorylation (pSer) status of Ubc9 in GSCs after inhibition of Plk1. Cells were treated with SBE 13 HCl (20 μ M) or DMSO for 12 hours. Cell lysates were subjected to immunoprecipitation with IgG or anti-Ubc9 antibodies followed by immunoblot with anti-pan-pSer antibodies. Inhibition of Plk1 reduced pSer of Ubc9 in GSCs.

(C) Co-Immunoprecipitation to determine the interaction between Ubc9 and Pin1 in GSCs after inhibition of Plk1. Cells were treated with SBE 13 HCl (20 μ M) or DMSO for 9 hours and harvested. MG132 (10 μ M) was added into cell culture 6 hours before harvest to obtain enough Pin1 proteins after Plk1 inhibition. Cell lysates were subjected to immunoprecipitation with IgG or anti-Ubc9 antibodies followed by immunoblot with anti-Pin1 antibodies. Inhibition of Plk1 did not affect the interaction between Ubc9 and Pin1 in GSCs.

Reviewer #1 (Minor comments): 7. The clinical relevance of RO-3306 is limited due to solubility issues. It is therefore not used in clinical trials. It is good to mention this in the discussion section of the manuscript.

Response: We thank the reviewer for the suggestion. We have added the description of the solubility issues of RO-3306 that limit its clinical application in the Discussion section (Line 593-595, Page 21) of the revised manuscript.

Reviewer #1 (Minor comments): 8. Please note that the correct term is 'Hoechst', not 'Hochest'.

Response: We apologize for the typo. We have corrected the term to Hoechst in the revised manuscript.

Reviewer #1 (Minor comments): 9. Typo Fig 6G: SUOM2

Response: We apologize for the typo. We have corrected the term to SUMO2/3 in the revised manuscript.

Reviewer #2

Reviewer #2: *In this manuscript, Zhu et al., aim to decipher the mechanism responsible for the high expression of the prolyl isomerase Pin1 that is associated with SUMO1 hypersumoylation in glioma stem cells (GSC). The authors, using classical biochemistry methodologies, follow a step by step approach to clearly demonstrate for the first time that USP34 interacts with Pin1 leading to its stabilization, induction of Ubc9 isomerization and SUMO1 hypersumoylation in GSCs. They began their study by showing for the that Pin1 interacts with USP34 leading to Pin1 deubiquitination and stabilization. Next, they studied how this interaction is regulated and the role of USP34 in GSC maintenance and in vivo glioma growth. The authors followed up by bridging USP34 role as Pin1 deubiquitinase with SUMO1 hypersumoylation in GSC. Next, the authors focused on Pin1 and its function as isomerase, demonstrating how Pin1 by promoting Ubc9 isomerization, facilitates Ubc9-SUMO1 thioester bond and SUMO1 hypersumoylation. Since CDK1 has been already shown to phosphorylates Ubc9, the authors tested the hypothesis that CDK1 might regulate the interaction of Ubc9 and pin1. Finally, the authors evaluated the co-inhibition of CDK1 and Pin1 with commercially available inhibitors in vitro on GSCs viability and maintenance and in vivo in a GBM model. The findings of the paper are bringing new light on the mechanism leading to SUMO1 hypersumoylation in GBM which is extremely relevant since it has been well documented that SUMO1 play a major role in GBM formation and growth. In addition this study may offer new therapeutic strategy in GBM treatment.*

Response: We thank the reviewer for the in-depth understanding of the manuscript.

Reviewer #2: Overall, this is a well-written, well-prepared manuscript with a lot of solid data provided. I have only one concern that maybe due to a mistake in the method writing. It seems based on the method section that all the IP for ubiquitination analysis were done in mild IP buffer (basically 1% NP-40). When analyzing the ubiquitination of any protein it is necessary to lysed in denaturing buffer to remove any interacting protein from the one studied. The lysate is then diluted to allow the IP. If as described in this paper and method, the IP was done in mild buffer, the authors cannot guarantee that the ubiquitination observed (for example in Fig1B, 1H) is really Pin1 ubiquitination and not one of the proteins it interacts with. Pin1 by nature as a peptidyl prolyl cis-trans isomerase interacts with numerous proteins including some for which Pin1 isomerization induced ubiquitination. Not IP'ing in denaturing buffer weakened the interpretation of the result.

Response: We totally understand the reviewer's concern about the potential bias in the immunoprecipitation because of the mild IP buffer. To disrupt the potential interaction between Pin1 and its ubiquitinated partners, cells were lysed in RIPA buffer (50 mM Tris-HCl pH7.4, 150 mM NaCl, 2 mM EDTA pH8.0, 1% NP-40, 0.1% SDS, 20 mM β -glycerophosphate, 1 mM Na_3VO_4 , 10 mM NaF, 1 mM PMSF, and protease inhibitor cocktail). The cell lysates were further diluted with PBS with a ratio of 1:5 for immunoprecipitation and ubiquitination analysis. With these settings, we repeated the experiments of Fig 1B and Fig 1H in the manuscript. The results showed that Pin1 had stronger ubiquitination in NSTCs relative to GSCs (Fig. R7A), and that disruption of USP34 elevated Pin1 ubiquitination in GSCs (Fig. R7B). Overall, the strength of the ubiquitination signals was similar in the experiments with mild buffer and the RIPA buffer (Fig. R7). Because the detection of the Pin1 ubiquitination was not altered with the mild lysis buffer (Fig. R7C and R7D, equal to Fig. 1B and 1H in the revised manuscript) relative to the strong lysis buffer (Fig. R7A and R7B), we would like to keep the immunoblot data as was.

Rebuttal Fig. R7

[Redacted]

Figure R7. (A) Immunoblot analysis of the poly-ubiquitination of endogenous Pin1 proteins in GSCs and NSTCs. GSCs and NSTCs were treated with MG132 (10 μ M) for 6 hours and then harvested with RIPA buffer for ubiquitination assays. Cell lysate was further diluted with PBS with a ratio of 1:5 and then subjected to immunoprecipitation with anti-Pin1 antibodies followed by immunoblot with anti-Ub antibodies. Strong poly-ubiquitination of Pin1 proteins was detected in NSTCs, but negligible poly-ubiquitination of Pin1 appeared in GSCs.

(B) Immunoblot analysis of the polyubiquitination of endogenous Pin1 proteins in GSCs expressing shUSP34. Cells were harvested 72 hours post-lentiviral infection. MG132 (10 μ M) was added to cell culture 6 hours before harvest for the ubiquitination assay. Cell lysate was further diluted with PBS with a ratio of 1:5 and then subjected to immunoprecipitation with anti-Pin1 antibodies followed by immunoblot with anti-Ub antibodies. Disruption of USP34 elevated polyubiquitination of Pin1 in GSCs.

(C) Immunoblot analysis of the poly-ubiquitination of endogenous Pin1 proteins in GSCs and NSTCs. GSCs and NSTCs were treated with MG132 (10 μ M) for 6 hours and then harvested with IP buffer for ubiquitination assays. Cell lysate was subjected to immunoprecipitation with anti-Pin1 antibodies followed by immunoblot with anti-Ub antibodies. Strong poly-ubiquitination of Pin1 proteins was detected in NSTCs, but negligible poly-ubiquitination of Pin1 appeared in GSCs.

(D) Immunoblot analysis of the poly-ubiquitination of endogenous Pin1 proteins in T4121 GSCs expressing shUSP34. Cells were harvested 72 hours post-lentiviral infection. MG132 (10 μ M) was added to cell culture 6 hours before harvest for the ubiquitination assay. Cells were lysed with IP buffer and then subjected to immunoprecipitation with anti-Pin1 antibodies

followed by immunoblot with anti-Ub antibodies. Disruption of USP34 elevated polyubiquitination of Pin1 in GSCs.

Reviewer #2 (Minor): *In the introduction line 142, the authors cited PIAS 1-4 as the E3 ligase. They should rephrase the sentence since PIAS1-4 are not the only SUMO E3-ligases, others such as Ran-BP2, TRIM.*

Response: We thank the reviewer for pointing out this important issue. We have rephrased the sentence and included other SUMO E3 ligases (Line 142-143, Page 5).

Reviewer #3

Reviewer #3: *In this manuscript, the authors studied the role of Pin1 in the hypersumoylation in glioma stem cells (GSC) and how this is essential for GSC stemness in vitro and tumor progression for in vivo PDX models. The results derived from this work are exciting since the authors depicted with a high level of detail how Pin1 gets stabilized in GSCs and how Pin1-mediated isomerization of Ubc9 promotes SUMO1 conjugation in GSC, promoting tumorigenesis. This work sheds some light on the understanding of the mechanisms of the high sumoylation in GBM. In terms of therapeutic approach, although targeting Pin1 and CSK1 are not novel strategies, the combination of both targeting the same pathways is interesting.*

In general, the results supported the conclusions and claims made by the authors, although I considered some revisions necessary. Also, the methodology used here adjusts to the questions that the authors aim to answer, and, in general, they are very well explained in the section Material and Methods.

The points that I consider need to be revised are the following.

Response: We thank the reviewer for the positive comments and the constructive suggestions. As advised by the reviewer, we have performed substantial additional experiments to further validate our main conclusions. The new data relevant to the main conclusions have been included in the revised manuscript. We believe that we have addressed the concerns raised by the reviewer with the revised manuscript.

Reviewer #3:1. *NSTC has not been defined. Please review all the acronyms and ensure that they have been defined for the first time they appear in the text.*

Response: We apologize for not defining the non-stem tumor cell (NSTC). We have carefully checked all the acronyms and made sure that they are defined when appeared in the text for the first time.

Reviewer #3: 2. *In general, the immunofluorescence images are very saturated, and it is difficult to distinguish any detail of the cells. Also, nuclei look larger than the cells in some images. In some cases, it is difficult to assume the claims of the authors due to the image quality. Also, it might be because of the conversion, but it looks like the contrast of those images has dramatically increased. Please, include better images.*

Response: We apologize for the unsatisfied quality of the immunofluorescence images. We have provided new images with better resolution in the revised manuscript. In addition, we cropped and enlarged a section of 3-5 cells for each new image to illustrate some details. These pictures clearly demonstrate the co-expression of USP34 and Pin1 in GBMs (Fig. R5A and R5B, equal to Fig. 1E and Supplementary Fig. 1C in the revised manuscript), and the co-staining of USP34 with SUMO1 but not SUMO2/3 (Fig. R5C and R5D, equal to Fig. 4D and Supplementary Fig. 3D and 3E in the revised manuscript). We believe that these new images have strengthened our main conclusions.

Rebuttal Fig. R5

[Redacted]

[Redacted]

Figure R5. (A and B) Representative images of immunofluorescent analysis of USP34 (red) and Pin1 (green) in human primary GBMs. Frozen sections of human GBMs were immunostained with antibodies against USP34 and Pin1, and counterstained with Hoechst to show nuclei (blue). The majority of USP34+ cells were positively stained for Pin1 in human GBMs. Scale bar, 20 μm .

(C and D) Representative images of immunofluorescent analyses of USP34 (red) and SUMO1 or SUMO2/3 (green) in human primary GBMs. Frozen sections of human GBMs were immunostained with antibodies against USP34 and SUMO1 or SUMO2/3, and counterstained with Hoechst to show nuclei (blue). The majority of SUMO1+ cells were positively stained for USP34 in human GBMs, whereas only about half of SUMO2/3+ cells had USP34 staining. Scale bar, 10 μm .

Reviewer #3: Also, In figure 4D, I would include NSTCs to see if SUMO1 correlates with USP34 in non-stem cells as well.

Response: We thank the reviewer for the suggestion. NSTCs contain different types of differentiated cancer cells and have no molecular marker in common. Therefore, we defined NSTCs as cells stained negative for stem cell markers such as SOX2. To determine if SUMO1 correlates with USP34 in NSTCs, we performed immunofluorescent staining on human primary GBMs with the antibodies against USP34, SUMO1, and SOX2. The results showed that whereas very few SOX2⁻ NSTC had SUMO1 staining, these NSTCs did not express USP34 (Fig. R8A and R8B, equals to Supplementary Fig. 3G and 3H in the revised manuscript), suggesting that potential mechanisms other than the USP34-Pin1-Ubc9 axis may regulate SUMO1-sumoylation in NSTCs. We have included these data in the revised manuscript as supplementary data related to Fig. 4D.

Rebuttal Fig. R8

[Redacted]

Figure R8. (A and B) Representative images (A) and statistical quantification (B) of immunofluorescent analyses of USP34 (red), SUMO1 (green), and SOX2 (white) in human primary GBMs. Frozen sections of human GBMs were immunostained with antibodies against USP34, SUMO1, and SOX2, and counterstained with Hoechst to show nuclei (blue). The percentage of SOX2⁺ GSCs stained with both USP34 and SUMO1 was quantified. Very few SOX2⁻ NSTCs that had SUMO1 staining showed no expression of USP34 (yellow arrows). Scale bar, 10 μ m. (n = 5 sections; mean \pm s.d.)

Reviewer #3: 3. In the point "Simultaneous inhibition of Pin1 and CDK1 suppresses SUMO1-modified sumoylation in GSCs and mitigates GBM growth", the authors suggest that apoptosis due to inhibition of Pin1 and CDK1 could be behind the slowdown in tumor growth. However, I am unsure whether the apoptosis shown in that image is enough. Additionally, how did the authors identify that the region imaged is actually the tumor?

Response: We thank the reviewer for raising the question. In fact, we found that the slowdown of tumor growth after inhibition of Pin1 and CDK1 could be ascribed to

multiple reasons. In addition to the elevated apoptosis, proliferation of tumor cells was markedly inhibited, as represented by the reduction of Ki67 staining (Fig. R9A and R9B, equal to Fig. 7G and 7H in the revised manuscript). GSCs are widely recognized as the most malignant population in GBMs. Immunofluorescent analyses showed that simultaneous inhibition of Pin1 and CDK1 most effectively reduced the OLIG2⁺ GSC proportion in GBM tumors (Fig. R9C and R9D, equal to Supplementary Fig. 5H and 5I in the revised manuscript), indicating that the combination therapy targeted GSCs to mitigate tumor malignancy. Therefore, elevated apoptosis, decreased proliferation, and impaired stemness altogether lead to the slowdown in tumor growth.

As to the approach for identification of the tumor region, we usually do it according to the size, intensity, and morphology of nuclei. Tumor cells have bigger nuclei than normal cells in mouse brains (Fig. R9E). In addition, tumor regions are crowded with tumor cells, whereas normal brain regions have scattered neural and glial cells, resulting in very different cell intensity (Fig. R9E). Moreover, nuclei of tumor cells are quite solid, whereas nuclei of normal mouse brain cells seem to have many tiny particles (Fig. R9E). Based on these features, we could easily distinguish tumor regions from normal brain regions in mice bearing GSC-derived GBM tumors.

Rebuttal Fig. R9

[Redacted]

Figure R9. (A and B) Immunofluorescent analysis (A) and statistical quantifications (B) of the cell proliferation marker Ki67 (green) and the apoptotic marker cleaved caspase-3 (red) in T4121 GSC-derived xenografts treated with sulfopin, RO3306, sulfopin plus RO3306, or DMSO control. Both sulfopin and RO3306 treatment suppressed cell proliferation but elevated apoptosis in GBM tumors. The combination treatment showed the strongest effects on cell proliferation and apoptosis. Scale bar, 40 μm (n = 5 tumors for each group; ***P < 0.001; mean \pm s.d.; two-tailed unpaired t-test).

(C and D) Immunofluorescent analysis (C) and statistical quantifications (D) of the stem cell marker OLIG2 in T387 GSC-derived xenografts treated with sulfopin, RO3306, sulfopin plus RO3306, or DMSO control. Both sulfopin and RO3306 treatment reduced the OLIG2+ GSC population, but the combination treatment resulted in the most significantly decrease of OLIG2 staining. Scale bar, 40 μm (n = 5 tumors for each group; ***P < 0.001; mean \pm s.d.; two-tailed unpaired t-test).

(E) Representative images of cell nuclei stained with Hoechst (blue) in GSC-derived xenografts. Cell intensity is much higher in tumor region than in normal region. Tumor cells have large and solid nuclei. Normal cells have small nuclei with many particles inside. Scale bar, 100 μm .

Reviewer #3: 4. The authors mention in the discussion that inhibition of global sumoylation could be potentially toxic for the cells. Therefore, targeting *Pin1*, which is the upstream regulator of *SUMO1*-conjugation could be a better therapeutic strategy for GBM. I wonder whether the treatment with Sulfopin induces toxicity in normal cells, such as neurons or primary astrocytes and whether they observed toxicity on mice during in vivo experiments.

Response: We thank the reviewer for this critical question. To explore the potential toxicity of sulfopin on normal cells, we stained brain sections from GBM-bearing mice treated with sulfopin or DMSO. The intensity and morphology of normal astrocytes and neurons in the normal brain regions in these GBM-bearing mice were investigated with immunofluorescent staining of the corresponding molecular markers GFAP and NeuN. The results showed that sulfopin treatment had no perceptible effects on the intensity of GFAP⁺ astrocytes or NeuN⁺ neurons in normal brain regions (Fig. R10A-D, equal to Supplementary Fig. 6A-6D in the revised manuscript). Nor did sulfopin treatment cause morphological changes in astrocytes or neurons (Fig. R10A-D, equal to Supplementary Fig. 6A-6D in the revised manuscript). Furthermore, we determined cell apoptosis in tumor and normal regions in brain sections from GBM-bearing mice. We detected no apparent apoptotic cells stained with cleaved caspase 3 in normal regions with or without sulfopin treatment, although sulfopin treatment markedly elevated cell apoptosis in tumor regions (Fig. R10E). Last but not least, we found that sulfopin treatment did not affect the weight loss of GBM-bearing mice, and another lab found that intraperitoneal injection of sulfopin caused no detectable pathologies in the post-mortem examination (*Nat Chem Biol.* 2021 Sep;17(9):954-963. doi: 10.1038/s41589-021-00786-7.). Taken together, sulfopin did

not induce toxicity in normal cells during in vivo experiments. We have included these data as supplementary data in the revised manuscript.

Rebuttal Fig. R10

[Redacted]

[Redacted]

Figure R10. (A and B) Representative images (A) and statistical quantification (B) of the GFAP⁺ astrocytes in normal brain regions in mice bearing GSC-derived GBM tumors after treatment with DMSO or sulfopin. Sulfopin treatment did not affect the intensity or the morphology of normal astrocytes. Scale bar, 20 μ m. (n = 5 brains for each group; mean \pm s.d.; two-tailed unpaired t-test).

(C and D) Representative images (C) and statistical quantification (D) of the NeuN⁺ neurons in normal brain regions in mice bearing GSC-derived GBM tumors after treatment with DMSO or sulfopin. Sulfopin treatment did not affect the intensity or the morphology of normal neurons. (n = 5 brains for each group; mean \pm s.d.; two-tailed unpaired t-test).

(E) Immunofluorescent analysis of the apoptotic marker cleaved caspase-3 in normal and tumor regions in brains from mice bearing GSC-derived xenografts after treatment with DMSO or sulfopin. Compared with the control treatment, sulfopin elevated cell apoptosis in tumor regions but not in normal brain regions. Scale bar, 40 μ m.

Reviewer #3: 5. *Although the authors intend to correct the bias associated with the different cell culture media (FBS for NSTC), I am wondering whether 12h in the same medium is enough. Do the authors have tried to assess the ubiquitination of Pin1 having GSCs in FBS or in NBM to see whether the FBS is affecting the protein stability and expression?*

Response: We thank the reviewer for this interesting question. As suggested by the reviewer, we sought to compare the ubiquitination of Pin1 in GSCs in FBS and in NBM. Because FBS would induce differentiation of GSCs, we initially determined how long would GSCs maintain their stemness in 10% FBS. Time-course analysis of the levels of the stem cell marker OLIG2 with immunoblot showed that 9-hour culture of GSCs in 10% FBS caused a severe loss of stemness, whereas 6-hour culture in FBS did not affect OLIG2 expression (Fig. R11A). We thus cultured GSCs in FBS or NBM for 6 hours and lysed the cells for the analysis of Pin1 ubiquitination. Ubiquitination assays showed that 6-hour culture in FBS had negligible effects on ubiquitination or stability of Pin1 in GSCs relative to those cultured in NBM (Fig. R11B). Therefore, different cell culture media did not introduce bias into the analysis of the ubiquitination of Pin1 in GSCs and NSTCs.

Rebuttal Fig. R11

[Redacted]

Figure R11. (A) Immunoblot analysis of OLIG2 protein levels in GSCs that have been cultured in the NBM supplemented with 10% FBS for the indicated durations. Expression of OLIG2 remained unchanged after 6-hour culture but declined after 9-hour culture in 10% FBS. (B) Immunoblot analysis of the poly-ubiquitination of endogenous Pin1 proteins in GSCs cultured in NBM or in NBM supplemented with 10% FBS for 6 hours. MG132 (10 μ M) was added to cell culture 6 hours before harvest for the ubiquitination assay. Cell lysate was subjected to immunoprecipitation with anti-Pin1 antibodies followed by immunoblot with anti-Ub antibodies. 6-hour culture in FBS had negligible effects on ubiquitination or stability of Pin1 in GSCs relative to those cultured in NBM.

Reviewer #3: 6. *The authors mention that most Pin1 inhibitors are unable to pass through BBB. Could they please provide any reference or experimental evidence that Sulfofin is actually able to reach the brain since the treatment is through intraperitoneal injections?*

Response: We thank the reviewer for this key question about the delivery of sulfofin into brains. Because there is no literature so far about the capacity of sulfofin to penetrate the blood-brain-barrier, we have to perform animal experiments to address the issue. For in vivo imaging of tumor growth, we are able to obtain strong bioluminescent signals in mouse brains 5 minutes post the intraperitoneal injection of luciferin. Therefore, we decided to harvest mouse brains 5 minutes post intraperitoneal injection of sulfofin (20 mg/kg) and utilize LC-MS/MS to detect the potential delivery of sulfofin into brains. The resultant chromatograms showed that a

peak representing sulfopin was detected in brains from mice treated with sulfopin but not the control mice (Fig. R12, equals to Supplementary Fig. 6E in the revised manuscript). These data support that sulfopin is able to reach the brain through intraperitoneal injection.

Rebuttal Fig. R12

[Redacted]

Figure R12. Representative LS-MS/MS chromatograms of the detection of sulfopin in mouse brains. Mouse brain tissue was ground and mixed with sulfopin to generate Standard (200 ng/mL) for LC-MS/MS analysis (left). For analysis of the delivery of sulfopin into brains, mice were treated with DMSO or sulfopin (20 mg/kg) through intraperitoneal injection and mouse brains were harvested 5 minutes post treatment. The brains were then ground and subjected to LS-MS/MS analysis. No sulfopin was detected in the brains from mice treated with DMSO (middle). Meanwhile, clear sulfopin signals were detected in the brains from mice treated with sulfopin (right).

REVIEWER COMMENTS

Reviewer #1 (Remarks to the Author):

The authors have addressed most of my concerns and the manuscript has significantly improved. Points 1, 2 and 5 require some follow up work.

1. I appreciate the efforts by the authors to identify SUMO1 target proteins, which is technically challenging. Since the authors use SUMO1-site independent methodology, I have some concerns about the results: a large fraction of identified proteins might not be bona fide SUMO1 target proteins. The authors need to compare SUMO1 mass spec results to known SUMO1 targets from the literature – what is the overlap? Since SUMO1 is mainly located in the nucleus, one would expect to find predominantly nuclear proteins. The authors need to verify whether the majority of identified SUMO1 target proteins are indeed nuclear. Because of the concerns, it is fine to keep the results in the rebuttal only and not include them in the revised manuscript.

2. The authors have now added mechanistic considerations in the discussion, however, the proper mechanism remains elusive. It is worthwhile to study the suggested mechanism in vitro. Strengthening the mechanistic aspect would enhance the impact of the manuscript.

3. The authors rule out roles for SENP1 and PIAS4.

4. The authors show that sumoylation of Pin1 on K6 and K63 is required for its stability.

5. The resolution of the microscopy has improved, but the images are still largely overexposed. Nevertheless, the images demonstrate the co-expression of USP34 and Pin1 in GBMs. Can the authors get access to better microscopes run by experienced microscopists to further improve the microscopy?

6. The authors have test whether Pin1 is also directly regulated by CDK1 and whether Ubc9 can be phosphorylated by Plk1. Whereas Ubc9 can indeed be phosphorylated by Plk1, this phosphorylation appeared to be irrelevant to the Pin1 catalyzed isomerization of Ubc9.

7. Properly addressed by the authors.

8. Properly addressed by the authors.

Concerning reviewer 2 - the point about the mild IP buffer. The authors now use RIPA buffer, but dilute this buffer 5x in PBS. Whereas RIPA buffer is good to use in this context, diluting it 5x with PBS is not good, because the result is still a mild buffer. I would like to ask that the authors use RIPA buffer not diluted with PBS to address this point.

Reviewer #3 (Remarks to the Author):

The authors have done an excellent job addressing all the comments of the reviewer and providing new data that supports their claims better. I have no additional comments.

Response to Reviewers' comments

Manuscript NCOMMS-23-12195A

Reviewer #1

Reviewer #1: *The authors have addressed most of my concerns and the manuscript has significantly improved. Points 1, 2 and 5 require some follow up work.*

Response: We thank the reviewer for the time and effort in reviewing our manuscript. We have performed additional experiments to address Points 1, 2 and 5. We believe that the revised manuscript has further clarified and consolidated our discoveries.

Reviewer #1: *1. I appreciate the efforts by the authors to identify SUMO1 target proteins, which is technically challenging. Since the authors use SUMO1-site independent methodology, I have some concerns about the results: a large fraction of identified proteins might not be bona fide SUMO1 target proteins. The authors need to compare SUMO1 mass spec results to known SUMO1 targets from the literature – what is the overlap? Since SUMO1 is mainly located in the nucleus, one would expect to find predominantly nuclear proteins. The authors need to verify whether the majority of identified SUMO1 target proteins are indeed nuclear. Because of the concerns, it is fine to keep the results in the rebuttal only and not include them in the revised manuscript.*

Response: We thank the reviewer for the constructive suggestions. We found a reference study (Identification of Small Ubiquitin-like Modifier Substrates with Diverse Functions Using the Xenopus Egg Extract System. *Molecular & Cellular Proteomics* 2014 Jul;13(7):1659-75.) using cobalt beads to pull-down substrate proteins conjugated with His-tagged SUMO proteins for mass spectrometry analysis. Of note, the sumoylated candidates were washed off the cobalt beads under denaturing conditions to get rid of the SUMO-interacting proteins. The reference study revealed many sumoylated proteins that led to follow-up studies from multiple labs, indicating the reliability and importance of their discoveries. By the way, although the reference study applied SUMO mutants and tried to identify the sites of sumoylation, they got a very low efficiency and obtained sumoylation sites for <5% of the candidate SUMO targets, highlighting the technical difficulties in such study.

We first searched for the protein localization information in the Uniprot website (<https://www.uniprot.org>) for SUMO1-modified targets identified in the reference study. Among a total of 214 proteins with localization information, most proteins (86 proteins) are localized in cytoplasm, and 59 proteins are localized in nucleus (Fig. R1A). We then looked into the sumoylated targets identified in our study. The results showed that among a total of 460 candidates with localization information in Uniprot, most proteins (215 proteins) are cytoplasmic proteins, and 69 proteins are nuclear proteins (Fig. R1A). Therefore, our mass spectrometry result is comparable to the data from the reference

study, and the largest portion of sumoylated proteins are cytoplasmic proteins, whereas a substantial fraction of sumoylated proteins are nuclear proteins.

When it comes to the reliability of the candidate SUMO targets identified by mass spectrometry, the reference study showed that among all the identified substrates, 9.8% were validated in other studies and 29.5% were identified by other high throughput reports (Fig. R1B, originally Fig. 4F from *Molecular & Cellular Proteomics* 2014 Jul;13(7):1659-75.). Our mass spectrometry analysis revealed 265 proteins that are preferentially sumoylated by SUMO1 in GSCs relative to NSTCs. We searched these proteins in the Pubmed website (<https://pubmed.ncbi.nlm.nih.gov/>) and found that sumoylation of 23 proteins (8.7%) had been validated by other labs (Fig. R1B). Due to the time constraint, we didn't search data from high throughput researches. Thus, our mass spectrometry analysis successfully identified a fraction of known sumoylated targets just like the reference study.

Interestingly, although the reference study aimed to investigate the sumoylation activity in *Xenopus* egg extracts, in which a high level of sumoylation was associated with sperm chromatin, when they performed gene ontology (GO) analysis of the identified SUMO substrates, the most represented biological process (39.5%) was linked to metabolic processes (Fig. R1C, originally Fig. 5A from *Molecular & Cellular Proteomics* 2014 Jul;13(7):1659-75.). Likewise, GO analysis showed that the targets preferentially sumoylated in GSCs relative to NSTCs were mostly enriched (40.5%) in metabolic processes (Fig. R1C). These comparable data underscore the importance of sumoylation in metabolic processes.

Taken together, we found that our mass spectrometry analysis should be generally reliable. However, further validation is required for individual candidates to ascertain their sumoylation, which would warrant future studies about the functions of sumoylation in GSCs.

Rebuttal Figure R1

[Redacted]

[Redacted]

Figure R1. (A) Pie charts showing the protein localization of sumoylated proteins identified by mass spectrometry in this study and the reference study. Protein localization information was obtained from the Uniprot website. The identified sumoylated proteins from both studies are mainly cytoplasmic proteins, whereas a substantial proportion of sumoylated proteins are found to localize in nucleus in both studies.

(B) Pie charts showing the fraction of total SUMO substrates that have been validated, identified or unknown prior to the studies. Among the 265 identified proteins that are preferentially sumoylated in GSCs relative to NSTCs in our study, SUMO conjugation have been validated on 23 proteins (listed below) by other studies published in Pubmed. For the SUMO substrates discovered by the reference study, 9.8% were validated by other labs, and 29.5% were identified by other high throughput screenings.

(C) Pie charts showing gene ontology analysis of the SUMO candidate proteins identified in our study and the reference study. The largest category is related to metabolic process.

Reviewer #1: 2. The authors have now added mechanistic considerations in the discussion, however, the proper mechanism remains elusive. It is worthwhile to study the suggested mechanism in vitro. Strengthening the mechanistic aspect would enhance the impact of the manuscript.

Response: We appreciate the reviewer for emphasizing the mechanistic study. Based on numerous data, we have drawn the conclusion that the Pin1-catalyzed isomerization enhances the affinity of Ubc9 to SUMO1, which does not necessarily affect the conjugation of SUMO2/3 to Ubc9. In order to provide direct evidence for this hypothesis, as suggested by the reviewer, we performed some in vitro experiments for mechanistic study. Ectopic Pin1 was introduced into 293T cells along with the wild type Ubc9 or the Ubc9-S71A mutant. Then the Ubc9 proteins were precipitated, and the conjugation of endogenous SUMO1 and SUMO2/3 modifiers to the wild type Ubc9 and the Ubc9-S71A mutant were determined by immunoblot. The results showed that overexpression of Pin1 markedly elevated the conjugation of SUMO1 to the wild type Ubc9 but not the Ubc9-S71A mutant, whereas the high levels of Ubc9-SUMO2/3 conjugates were not affected by neither Pin1 overexpression nor mutation of the pS71-Pro motif (Fig. R2, equals to Supplementary Fig. 4I), indicating that Pin1-catalyzed isomerization of Ubc9 increases its affinity to SUMO1 without antagonizing SUMO2/3 conjugation in general. When it comes to the potential structural change upon isomerization that regulates the affinity of Ubc9 to SUMO1 and SUMO2/3, techniques such as cryo-electron microscopy may be necessary to address the issue. However, this is beyond our capacity and the scope of this study.

We believe that these new data have further clarify the mechanistic aspect of the influence of Pin1 on SUMO1 sumoylation. These data have been included in the revised manuscript as Supplementary Fig. 4I.

Rebuttal Figure R2

[Redacted]

Figure R2. Immunoblot analysis of the Ubc9-SUMO1 thioester (Ubc9~SUMO1) and Ubc9-SUMO2/3 thioester (Ubc9~SUMO2/3) formation in 293T cells overexpressing ectopic Pin1 along with wild type or the Ubc9-S71A mutant. Myc-tagged Ubc9 and flag-tagged Pin1 were overexpressed in 293T cells. Cell lysates were subjected to immunoprecipitation with anti-myc antibodies. The Ubc9-SUMO1 and Ubc9-SUMO2/3 conjugates were detected by immunoblot

with anti-SUMO1 and anti-SUMO2/3 antibodies, respectively. DTT and β -ME were added into protein input but not the immunoprecipitation products. Overexpression of Pin1 markedly elevated the conjugation of SUMO1 to the wild type Ubc9 but not the Ubc9-S71A mutant. Meanwhile, Pin1 overexpression showed negligible effects on the amount of the Ubc9-SUMO2/3 conjugates in any samples.

Reviewer #1: 3. *The authors rule out roles for SENP1 and PIAS4.*

Response: We thank the reviewer for agreeing with us that SENP1 and PIAS4 are not regulated by Pin1.

Reviewer #1: 4. *The authors show that sumoylation of Pin1 on K6 and K63 is required for its stability.*

Response: We thank the reviewer for acknowledging our discovery that sumoylation of Pin1 on K6 and K63 is required for its stability.

Reviewer #1: 5. *The resolution of the microscopy has improved, but the images are still largely overexposed. Nevertheless, the images demonstrate the co-expression of USP34 and Pin1 in GBMs. Can the authors get access to better microscopes run by experienced microscopists to further improve the microscopy?*

Response: We apologize for the unsatisfied quality of the microscopy pictures. We have consulted an experienced microscopists and replaced the overexposed pictures in the revised manuscript. In detail, we have replaced Supplementary Fig. 1C (Fig. R3A), Fig. 4D (Fig. R3B), Supplementary Fig. 3D (Fig. R3C), and Supplementary Fig. 3E (Fig. R3D). We believe that these new images have strengthened our main conclusions.

Rebuttal Figure R3

[Redacted]

Figure R3. (A) Representative images of immunofluorescent analysis of USP34 (red) and Pin1 (green) in human primary GBMs. Frozen sections of human GBMs were immunostained with antibodies against USP34 and Pin1, and counterstained with Hoechst to show nuclei (blue). The majority of USP34+ cells were positively stained for Pin1 in human GBMs. Scale bar, 20 μm .

(B) Representative images of immunofluorescent analyses of USP34 (red) and SUMO1 or SUMO2/3 (green) in human primary GBMs. Frozen sections of human GBMs were immunostained with antibodies against USP34 and SUMO1 or SUMO2/3, and counterstained with Hoechst to show nuclei (blue). Scale bar, 20 μm .

(C and D) Representative images of immunofluorescent analyses of USP34 (red) and SUMO1 (C) or SUMO2/3 (D) (green) in human primary GBMs. Frozen sections of human GBMs were immunostained with antibodies against USP34 and SUMO1 or SUMO2/3, and counterstained

with Hoechst to show nuclei (blue). The majority of SUMO1+ cells were positively stained for USP34 in human GBMs, whereas some SUMO2/3+ cells had USP34 staining. Scale bar, 20 μ m.

Reviewer #1: 6. *The authors have test whether Pin1 is also directly regulated by CDK1 and whether Ubc9 can be phosphorylated by Plk1. Whereas Ubc9 can indeed be phosphorylated by Plk1, this phosphorylation appeared to be irrelevant to the Pin1 catalyzed isomerization of Ubc9.*

Response: We thank the reviewer for agreeing with us that Plk1-mediated phosphorylation of Ubc9 is likely irrelevant to the Pin1-catalyzed isomerization of Ubc9.

Reviewer #1: 7. and 8. *Properly addressed by the authors.*

Response: We thank the reviewer for the endorsement.

Reviewer #1: *Concerning reviewer 2 - the point about the mild IP buffer. The authors now use RIPA buffer, but dilute this buffer 5x in PBS. Whereas RIPA buffer is good to use in this context, diluting it 5x with PBS is not good, because the result is still a mild buffer. I would like to ask that the authors use RIPA buffer not diluted with PBS to address this point.*

Response: We thank the reviewer for the suggestion. As suggested by the reviewer, to clarify that the ubiquitin signals were from Pin1 rather than Pin1-interacting proteins, we applied RIPA buffer throughout the ubiquitination assays to repeat the experiments of Fig. 1B in the manuscript. The results recapitulated previous observations that Pin1 had stronger ubiquitination in NSTCs relative to GSCs (Fig. R4). RIPA buffer is a strong buffer that may affect the binding of antibodies to antigens and thus is rarely used for immunoprecipitation. Therefore, we would like to keep the immunoblot data using mild buffer to avoid confusion.

Rebuttal Figure R4

[Redacted]

Figure R4. Immunoblot analysis of the poly-ubiquitination of endogenous Pin1 proteins in GSCs and NSTCs. GSCs and NSTCs were treated with MG132 (10 μ M) for 6 hours and then

harvested with RIPA buffer for ubiquitination assays. Cell lysate was subjected to immunoprecipitation with anti-Pin1 antibodies followed by immunoblot with anti-Ub antibodies. RIPA buffer was used as the buffer for immunoprecipitation. Strong poly-ubiquitination of Pin1 proteins was detected in NSTCs, but much weaker poly-ubiquitination of Pin1 appeared in GSCs.

Reviewer #3

***Reviewer #3:** The authors have done an excellent job addressing all the comments of the reviewer and providing new data that supports their claims better. I have no additional comments.*

Response: We thank the reviewer for the time and effort in reviewing our manuscript.#

REVIEWERS' COMMENTS

Reviewer #1 (Remarks to the Author):

The authors have addressed my remaining concerns, thereby strengthening the manuscript.

The microscopy experiments have improved, but images are still overexposed unfortunately and have rather poor resolution. But I do understand if they could not be improved further, in case the authors don't have access to better equipment.

Concerning the mass spectrometry analysis of SUMO1 targets, having a majority of cytoplasmic proteins is not consistent with the predominant nuclear location of SUMO1. My concern about potential false positive SUMO1 target proteins thus remains. Because of the concerns, it is fine to keep these results in the rebuttal only and not include them in the revised manuscript.

Response to Reviewers' comments

Manuscript NCOMMS-23-12195B

Reviewer #1

Reviewer #1: The authors have addressed my remaining concerns, thereby strengthening the manuscript.

Response: We thank the reviewer for the time and effort in reviewing our manuscript.

Reviewer #1: 1. The microscopy experiments have improved, but images are still overexposed unfortunately and have rather poor resolution. But I do understand if they could not be improved further, in case the authors don't have access to better equipment.

Response: We are sorry that the quality of the microscopy pictures still does not meet the reviewer's expectation. We thank the reviewer for the understanding.

Reviewer #1: 2. Concerning the mass spectrometry analysis of SUMO1 targets, having a majority of cytoplasmic proteins is not consistent with the predominant nuclear location of SUMO1. My concern about potential false positive SUMO1 target proteins thus remains. Because of the concerns, it is fine to keep these results in the rebuttal only and not include them in the revised manuscript.

Response: We totally understand the reviewer's concern. The immunofluorescent staining demonstrated that SUMO1 signals were enriched in nucleus, suggesting that many nuclear proteins may have heavy SUMO1-modified sumoylation. However, many cytoplasmic proteins of low abundance may be lightly sumoylated. Therefore, in terms of proteins types with SUMO1-modification, it is possible that the majority of SUMO1-sumoylated substrates may be cytoplasmic proteins. Anyway, further validation is required for individual candidates to ascertain their sumoylation, especially in the study of these cytoplasmic sumoylated proteins identified by mass spectrometry. These results will not be included in the final version of our manuscript.